# Estimating Neural Representation Alignment from Sparsely Sampled Inputs and Features

**Chanwoo Chun** [1]
Weill Cornell Medicine
Flatiron Institute

**Abdulkadir Canatar** [1]
Flatiron Institute

**SueYeon Chung**
Flatiron Institute
New York University

**Daniel D. Lee**
Cornell Tech
Flatiron Institute

## Abstract

**In both artificial and biological systems, the centered kernel alignment (CKA) has become a widely used tool for quantifying neural representation similarity. While current CKA estimators typically correct for the effects of finite stimuli sampling, the effects of sampling a subset of neurons are overlooked, introducing notable bias in standard experimental scenarios. Here, we provide a theoretical analysis showing how this bias is affected by the representation geometry. We then introduce a novel estimator that corrects the bias for both input and feature sampling. We use our method for evaluating both brain-to-brain and model-to-brain alignments and show that it delivers reliable comparisons even with very sparsely sampled neurons. We perform within-animal and across-animal comparisons on electrophysiological data from visual cortical areas V1, V4, and IT, and use these as benchmarks to evaluate model-to-brain alignment. We also apply our method to reveal how object representations become progressively disentangled across layers in both biological and artificial systems. These findings underscore the importance of correcting feature-sampling biases in CKA and demonstrate that our bias-corrected estimator provides a more faithful measure of representation alignment. The improved estimates increase our understanding of how neural activity is structured across both biological and artificial systems.**

**Keywords:** CKA; Representation Alignment; Disentanglement; Estimation

## Introduction

Over the past decade, the concept of *representation similarity* has emerged as a powerful framework for comparing complex neural and computational systems (Kriegeskorte et al., 2008; Kriegeskorte & Kievit, 2013). One of the most popular tools is *centered kernel alignment* (CKA), a measure originally adapted from kernel-based independence metrics but now widely employed across fields such as machine learning, neuroscience, and cognitive science. In machine learning, CKA has become the de facto standard for quantifying how similarly different layers—or even entirely different architectures—encode the same input data (Kornblith et al., 2019). In neuroscience, CKA has emerged as a core analytical tool to assess whether neural populations, either within or across brain regions and species, produce similar activity patterns in response to identical stimuli (Yamins & DiCarlo, 2016; Schrimpf et al., 2018). Its popularity

arises from two advantages: (1) CKA is invariant to orthogonal transformations, making it robust to small perturbations in feature space, and (2) it normalizes for overall variation in activity levels, facilitating meaningful comparisons based on alignment.

Despite these strengths, there is a growing consensus that conventional CKA estimators overlook a critical limitation in many experimental settings, that the sampling of only a subset of "features", as with experimental recordings, can introduce a systematic bias. This is particularly significant in neuroscience, where only a fraction of the total neural population is recorded. Existing CKA estimators often assume that if enough data points (e.g., stimuli or input images) are provided, the measure becomes reliable. Yet this assumption ignores the additional requirement for sufficiently large samples along the "feature" dimension. Consequently, researchers risk drawing misleading conclusions about the alignment of brain regions and neural network layers (Murphy et al., 2024; Cloos et al., 2024; Han et al., 2023; Sucholutsky et al., 2023).

In this work, we address this pressing concern. First, we show analytically how the geometry of high-dimensional representations contributes to spurious underestimation of similarity when only limited neuronal or model "units" are observed. Second, building on these insights, we introduce a novel estimator designed to remain consistent even with limited feature samples. By systematically correcting for finite-sample effects in both inputs and features, our method offers a more faithful gauge of representation alignment.

We demonstrate the real-world impact of our estimator through analyses of convolutional neural networks and multi-electrode electrophysiological recordings in visual cortices V1, V4, and IT (Papale et al., 2025). Our new estimator enables more accurate model-to-brain comparisons, revealing alignment trends that are otherwise corrupted by sampling biases. Beyond model-to-brain alignment, we show that the improved CKA estimator illuminates how object-category representations become increasingly disentangled along the primate ventral stream, similar to the observations in deep neural networks. Taken together, these results show that accounting for feature sampling is indispensable for robust representation analysis. Our work thus not only strengthens the theoretical foundations of CKA but also expands its practical utility for probing neural and computational representations.

## Problem Statement

There are many scenarios where accurate quantification of the similarity between representations is of central interest (Figure 1). For example, for a given stimulus set, one may

---

[1] These authors contributed equally to this work.

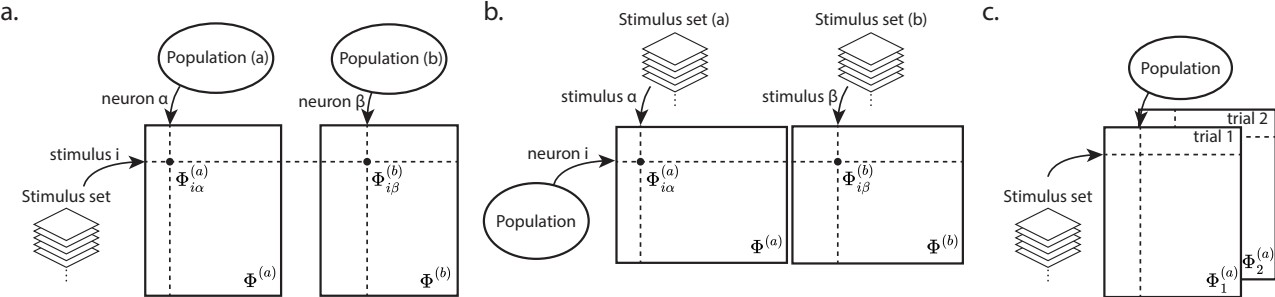

Figure 1: CKA can be considered in multiple different problem setups. Each rectangle represents a measurement matrix, and the dotted lines indicate specific rows and column indices. (a) Comparing the representations of two neural populations for one stimulus set. (b) Comparing the representations of two stimulus sets in one neural population. (c) Comparing the representations of a single population on a single stimulus set across two trials.

want to compare the representations of the brain and a neural network model. As a baseline for brain-to-model similarity, it is also useful to measure brain-to-brain similarities across two individual animals or across trials in one animal. In a separate scenario, one may be interested in comparing the neural representations of two stimulus sets, e.g. two image categories, in a single brain region of one individual animal. In all these setups, CKA can be used to quantify the similarities.

Suppose one measures two separate neural representation matrices, $\mathbf{X}$ and $\mathbf{Y}$, where each row corresponds to a distinct stimulus and each column represents a recorded neuron. Here, assume the data has been preprocessed so that each neuron has zero mean across all the recorded stimuli. Typically, in its most intuitive form, the CKA is then computed as

$$\frac{\text{tr}(\mathbf{X}\mathbf{X}^\top\mathbf{Y}\mathbf{Y}^\top)}{\sqrt{\text{tr}\left(\left(\mathbf{X}\mathbf{X}^\top\right)^2\right)\text{tr}\left(\left(\mathbf{Y}\mathbf{Y}^\top\right)^2\right)}} \quad (1)$$

In a hypothetical scenario where one has access to all neurons and stimuli, CKA takes the value $1$ when the representations are perfectly aligned and $0$ when they span orthogonal subspaces. The normalization factor in the denominator keeps CKA invariant to the scaling of the representations.

In reality, the representation matrices are actually sampled submatrices of an unobserved larger matrix. Due to practical limitations, we often do not get to observe all neurons in a brain region, and it is not possible to present all possible stimuli from a stimulus set, e.g. natural images. Due to these sampling effects, computing the naive CKA from the measurement matrices is heavily biased. Currently, the estimator derived from Song et al. (2012) is widely used in the literature to correct the bias contributed by stimulus sampling. In this paper, we show how to correct the bias in the CKA estimator when both the stimuli and neurons are sampled from the underlying large matrix.

## Our Contribution

We make the following contributions in this paper:

- We find that the popular CKA estimator based on Song et al. (2012) can take arbitrarily small values under finite sampling of both the stimuli and neurons, even when the underlying representations are perfectly aligned. Through theoretical analysis, we show that the bias increases with the intrinsic dimensionality of the representation.

- To address this issue, we develop a more general CKA estimator that corrects the bias contributed by both the stimulus and neuron sampling. We demonstrate its reliability in both synthetic and neurophysiological data.

- We demonstrate our estimator enables a novel application of CKA that quantifies the representation disentanglement in the brain from real data.

## Definitions

### Centered Kernel Alignment

A neural population $\boldsymbol{\phi}(x)$ defines a representation of stimuli $x \in \mathcal{X}$ with a large number of neurons. The associated *centered kernel function* $k(x,x')$ measures the similarity of two stimuli in the representation space of $\boldsymbol{\phi}$ and is defined as:

$$k(x,x') := \left\langle \boldsymbol{\phi}(x) - \mathbb{E}[\boldsymbol{\phi}] \,,\, \boldsymbol{\phi}(x') - \mathbb{E}[\boldsymbol{\phi}] \right\rangle, \quad (2)$$

where $\langle \cdot, \cdot \rangle$ denotes an appropriate inner product. Here, $\mathbb{E}[\boldsymbol{\phi}]$ denotes the expectation of $\boldsymbol{\phi}$ over the stimulus space $\mathcal{X}$ and is used to center the population activity.

For two distinct neural populations $\boldsymbol{\phi}^{(a)}(x)$ and $\boldsymbol{\phi}^{(b)}(x)$, we measure their similarity using the Hilbert-Schmidt Independence Criterion (HSIC), a popular metric in machine learning (Gretton et al., 2005). HSIC compares two populations based on the correlations of their kernel functions $k^{(a)}$ and $k^{(b)}$, and yields, what we call an $\mathcal{H}$-value:

$$\mathcal{H}(k^{(a)}, k^{(b)}) := \mathbb{E}_{x,x'}[k^{(a)}(x,x')k^{(b)}(x,x')]. \quad (3)$$

A low $\mathcal{H}$-value indicates less similarity and is zero if and only if the populations $\boldsymbol{\phi}^{(a)}$ and $\boldsymbol{\phi}^{(b)}$ vary independently.

The centered kernel alignment (CKA) is essentially the normalized version of HSIC ([Cortes et al., 2012](#); [Kornblith et al., 2019](#)) and is defined as

$$\text{CKA}(k^{(a)}, k^{(b)}) := \frac{\mathcal{H}(k^{(a)}, k^{(b)})}{\sqrt{\mathcal{H}(k^{(a)}, k^{(a)})\mathcal{H}(k^{(b)}, k^{(b)})}}. \quad (4)$$

CKA is normalized to the interval $[0, 1]$ and invariant to the overall magnitude of population activations.

**Measurement matrix**

Greek indices $(\alpha, \beta, \cdots)$ denote neurons; Latin indices $(i, j, \cdots)$ denote stimuli, and Latin indices $(a, b, \cdots)$ denote distinct populations. We use the letter $Q$ for the number of neurons and $P$ for the number of stimuli. Quantities with a hat ($\hat{\ }$) indicate empirical estimates.

For each population $\boldsymbol{\phi}^{(a)}$, we observe $Q_a$ neurons $\boldsymbol{\phi}_\alpha(x)$ for $\alpha = 1, \cdots, Q_a$. Each neuron is measured on the same set of stimuli $\{x_i\}$ for $i = 1, \cdots, P$, where $P$ is the number of stimuli. The corresponding measurement matrix $\boldsymbol{\Phi}^{(a)} \in \mathbb{R}^{P \times Q_a}$ with elements $\boldsymbol{\Phi}_{i\alpha}^{(a)} = \boldsymbol{\phi}_\alpha^{(a)}(x_i)$ denotes the response of each neuron to stimulus $x_i$.

We define the empirical *uncentered* kernels (also called Gram matrices) by

$$\mathbf{K}^{(a)} = \frac{1}{Q_a}\boldsymbol{\Phi}^{(a)}\boldsymbol{\Phi}^{(a)\top},$$

and the empirical centered kernels defined in Equation (2) by

$$\bar{\mathbf{K}}^{(a)} = \mathbf{H}\mathbf{K}^{(a)}\mathbf{H}, \quad \mathbf{H} = \mathbf{I} - \frac{1}{P}\mathbf{1}\mathbf{1}^\top, \quad (5)$$

where $\mathbf{H}$ is the centering matrix.

## Existing CKA estimators

With these definitions, we now construct estimators of $\mathcal{H}$-value from finite data. Here, we convey the overall ideas and provide detailed analyses in Appendix A.

**Naive Estimator**

Following the definition of $\mathcal{H}$-value in Equation (3), the naive estimator is

$$\widehat{\mathcal{H}_0}(k^{(a)}, k^{(b)}) = \frac{1}{P^2}\text{tr}\left(\bar{\mathbf{K}}^{(a)}\bar{\mathbf{K}}^{(b)}\right) \quad (6)$$

To see why this estimator is heavily biased, let us consider a simpler case where the original kernel is already centered such that $\mathbf{K} = \bar{\mathbf{K}}$. Then, the above can be simply rewritten as

$$\widehat{\mathcal{H}_0}(k^{(a)}, k^{(b)}) = \frac{1}{P^2 Q^2}\sum_{i,j}\sum_{\alpha,\beta} v_{ijji}^{\alpha\beta} \quad \text{where} \quad (7)$$

$$v_{ijlr}^{\alpha\beta} := \Phi_{i\alpha}^{(a)}\Phi_{j\alpha}^{(a)}\Phi_{l\beta}^{(b)}\Phi_{r\beta}^{(b)} \quad (8)$$

We need to compute the expected value of $\widehat{\mathcal{H}_0}(k^{(a)}, k^{(b)})$ (averaged over the random sampling of neurons and stimuli) to

see its bias. To compute the expected value, we first need to decompose the above sum as:

$$P^2 Q^2 \left\langle \widehat{\mathcal{H}_0}(k^{(a)}, k^{(b)}) \right\rangle_\Phi =$$
$$\sum_{i \neq j}\sum_{\alpha \neq \beta}\langle v_{ijji}^{\alpha\beta}\rangle_\Phi + \sum_{i=j}\sum_{\alpha \neq \beta}\langle v_{iiii}^{\alpha\beta}\rangle_\Phi$$
$$+ \sum_{i \neq j}\sum_{\alpha=\beta}\langle v_{ijji}^{\alpha\alpha}\rangle_\Phi + \sum_{i=j}\sum_{\alpha=\beta}\langle v_{iiii}^{\alpha\alpha}\rangle_\Phi \quad (9)$$

Note that the summand of the first term $\langle v_{ijij}^{\alpha\beta}\rangle_\Phi$ is exactly $\mathcal{H}(k^{(a)}, k^{(b)})$, the quantity we want to recover. On the other hand, the other sums with overlapping indices contribute to the bias. In general, unbiased estimates are obtained by averaging over all indices where each stimulus is used at most once ([Hoeffding, 1948](#)).

Now returning to Equation (6), in general, if we assume the original kernel is not centered, Equation (6) expands to

$$\widehat{\mathcal{H}_0}(k^{(a)}, k^{(b)}) = \langle v_{ijij}^{\alpha\beta}\rangle_{\text{naive}} - 2\langle v_{ijjl}^{\alpha\beta}\rangle_{\text{naive}} + \langle v_{ijlr}^{\alpha\beta}\rangle_{\text{naive}} \quad (10)$$

where the notation $\langle\cdot\rangle_{\text{naive}}$ is equivalent to averaging over all indices in the bracket, e.g. $\langle y_{ij}^{\alpha\beta}\rangle_{\text{naive}} = \frac{1}{Z}\sum_{i,j,\alpha,\beta} y_{ij}^{\alpha\beta}$ with $Z$ being the number of summands.

**Stimulus-Corrected Estimator**

The estimator developed by [Song et al. (2012)](#) corrects for the bias contributed by stimulus sampling. In this case, the term where two independent stimulus indices $(i, j)$ coincide would contribute to the bias. Therefore, [Song et al. (2012)](#) simply sums over disconnected $(i, j)$ indices:

$$\widehat{\mathcal{H}_S}(k^{(a)}, k^{(b)}) = \langle v_{ijji}^{\alpha\beta}\rangle_{\text{stim}} - 2\langle v_{ijjl}^{\alpha\beta}\rangle_{\text{stim}} + \langle v_{ijlr}^{\alpha\beta}\rangle_{\text{stim}} \quad (11)$$

where the notation $\langle\cdot\rangle_{\text{stim}}$ is equivalent to averaging over all neuron indices and disconnected stimulus indices in the bracket, e.g. $\langle y_{ij}^{\alpha\beta}\rangle_{\text{stim}} = \frac{1}{Z}\sum_{i \neq j}\sum_{\alpha,\beta} y_{ij}^{\alpha\beta}$.

The sample corrected estimator for CKA, denoted by $\widehat{\text{CKA}_S}(k^{(a)}, k^{(b)})$, is obtained by replacing $\widehat{\mathcal{H}_S}$ in Equation (4). This is the current version of CKA that is used both in deep learning ([Nguyen et al., 2021](#); [Raghu et al., 2021](#); [Davari et al., 2022](#)) and neuroscience ([Murphy et al., 2024](#)).

Note that this estimator removes the bias due to coinciding stimulus indices. However, the inputs to $\widehat{\mathcal{H}_S}(k^{(a)}, k^{(b)})$ are kernels which involve a sum over neurons. Therefore, if its inputs have *shared neurons* as in the case of the denominator of CKA, a similar bias discussed in Equation (9) occurs, but for neuron indices. Therefore, assuming that the populations $\boldsymbol{\phi}^{(a)}$ and $\boldsymbol{\phi}^{(b)}$ have independent neurons, the numerator of $\widehat{\text{CKA}_S}(k^{(a)}, k^{(b)})$ remains unbiased, but its denominator is biased due to finite neuron sampling effects (See Appendix B.2, Equation (S14)).

**Bias of the Existing Estimator**

We find that the representation geometry affects the bias in the widely-used stimulus-corrected estimator $\widehat{\text{CKA}_S}$. Assuming

the variance of the $\widehat{\mathcal{H}}_S$ estimate is negligible, and the activation variance is normalized neuron-wise, the bias of $\widehat{\text{CKA}}_S$ can be approximated as

$$\mathbb{E}\left[\widehat{\text{CKA}}_S(k^{(a)},k^{(b)})\right] - \text{CKA}(k^{(a)},k^{(b)}) \approx$$

$$\left[\frac{1}{\sqrt{\left(1+\frac{\gamma_a-1}{Q_a}\right)\left(1+\frac{\gamma_b-1}{Q_b}\right)}} - 1\right]\text{CKA}(k^{(a)},k^{(b)}) \quad (12)$$

where $\gamma_a$ and $\gamma_b$ are the intrinsic dimensionalities of the two underlying representations, quantified by the participation ratio of the eigenvalues $\{\lambda_i\}$ of $\bar{\mathbf{K}}$ in the infinite samples limit:

$$\gamma = \frac{(\sum_i \lambda_i)^2}{\sum_i \lambda_i^2}.$$

From Equation (12), we make the following observations:

- $\widehat{\text{CKA}}_S$ always underestimates the true CKA on average.

- The scale by which $\widehat{\text{CKA}}_S$ underestimates is independent of the alignment but only dependent on the intrinsic dimensionalities and neuron sample sizes.

- Having larger intrinsic dimensionalities contributes to the greater underestimation of CKA and requires more neuron samples to mitigate the bias.

Therefore, if intrinsic dimensionality is much greater than the number of feature samples, $\gamma \gg Q$, $\widehat{\text{CKA}}_S$ can take an arbitrarily small value even when two representations are perfectly aligned. On the other hand, our estimator is not affected by this issue. For more details, including the bias of $\widehat{\text{CKA}}_0$, see Appendix B.2.

### Stimulus-Neuron-Corrected Estimator

Unlike the previous two estimators, an unbiased estimator correcting for both finite stimulus and neuron sampling must be a function of populations $\boldsymbol{\phi}^{(a)}$ rather than their kernels.

In this paper, we develop a novel estimator $\widehat{\mathcal{H}}_C$ that corrects for finite neuron sampling effects when two populations have correlated, e.g., identical neurons. This can be done by summing over disconnected stimulus $(i,j)$ and neuron $(\alpha,\beta)$ indices:

$$\widehat{\mathcal{H}}_C(k^{(a)},k^{(b)}) = \langle v_{ijji}^{\alpha\beta}\rangle_{\text{both}} - 2\langle v_{ijjl}^{\alpha\beta}\rangle_{\text{both}} + \langle v_{ijlr}^{\alpha\beta}\rangle_{\text{both}} \quad (13)$$

where the notation $\langle\cdot\rangle_{\text{both}}$ is equivalent to averaging over disconnected neuron indices and disconnected stimulus indices in the bracket, e.g. $\langle x_{ij}^{\alpha\beta}\rangle_{\text{both}} = \frac{1}{Z}\sum_{i\neq j}\sum_{\alpha\neq\beta}x_{ij}^{\alpha\beta}$.

To implement this estimator in practice, each term needs to be expanded into a linear combination of the regular summations that do not sum over disconnected indices. For example, $\sum_{i\neq j}x_{ij}$ can be expanded as $\sum_{i,j}x_{ij} - \sum_i x_{ii}$, which is much easier to implement and more efficient in practice. When there are

more than two indices, the expansion becomes non-trivial. We leave the derivation of the fully expanded form of Equation (13) to Appendix A. An implementation is provided in Appendix D.

Finally, we define our unbiased CKA estimator as

$$\widehat{\text{CKA}}_C\left(\boldsymbol{\phi}^{(a)},\boldsymbol{\phi}^{(b)}\right) = \frac{\widehat{\mathcal{H}}_C(\boldsymbol{\phi}^{(a)},\boldsymbol{\phi}^{(b)})}{\sqrt{\widehat{\mathcal{H}}_C(\boldsymbol{\phi}^{(a)},\boldsymbol{\phi}^{(a)})\widehat{\mathcal{H}}_C(\boldsymbol{\phi}^{(b)},\boldsymbol{\phi}^{(b)})}}. \quad (14)$$

Several remarks are in order:

- Our estimator $\widehat{\mathcal{H}}_C$ reduces to the stimulus-corrected estimator $\widehat{\mathcal{H}}_S$ of Song et al. (2012) when two populations $\boldsymbol{\phi}^{(a)}$ and $\boldsymbol{\phi}^{(b)}$ have independent neurons, generalizing previous results.

- When two distinct populations compared, our CKA estimator $\widehat{\text{CKA}}_C$ corrects the bias in the denominator of $\widehat{\text{CKA}}_S$ and their numerators remain identical. However, in certain cases, e.g. trial-to-trial similarity between two measurements of a single population, the numerator of $\widehat{\text{CKA}}_S$ also becomes biased.

- Note that while the $\mathcal{H}$-estimator we derive is unbiased, the proposed CKA-estimator is still biased due to non-linear operations (multiplication, square-root) involving $\widehat{\mathcal{H}}_C$. Nevertheless, our CKA estimator is the least biased among the available estimators. Full analysis on this matter can be found in Appendix B.3. Also, as shown in the rest of the paper, we empirically find that the effect of this bias is much smaller compared to the effect of bias due to finite neuron sampling. One can also mitigate this bias by empirically averaging $\widehat{\mathcal{H}}$ over multiple trials and using that to estimate CKA. If $N$ estimates of $\widehat{\mathcal{H}}$ are available from independent trials, then this bias falls like $O\left(1/N\right)$ (Appendix B.3). We will denote the CKA estimator derived from the empirical average of $N$ number of $\widehat{\mathcal{H}}$ by $\widehat{\text{CKA}}^N$.

### Linear example

Next, we numerically test all three estimators on a simple synthetic dataset with a known CKA value. We consider $d-$dimensional stimuli which are drawn from the distribution $\mathbf{x} \sim \mathcal{N}(0,\mathbf{I}_d)$. We define a linear population of the form $\boldsymbol{\phi}^{(a)}(\mathbf{x}) = \mathbf{x}^\top\mathbf{w}^{(a)}$, where each weight is drawn from the distribution $\mathbf{w}^{(a)} \sim \mathcal{N}(0,\boldsymbol{\Sigma}_a)$ and corresponds to a single neuron. Similarly, we define the population $\boldsymbol{\phi}^{(b)}(\mathbf{x}) = \mathbf{x}^\top\mathbf{w}^{(b)}$ where $\mathbf{w}^{(b)} \sim \mathcal{N}(0,\boldsymbol{\Sigma}_b)$. Since the entire stimuli and population distributions are known, the true CKA in Equation (4) can be evaluated exactly:

$$\text{CKA} = \frac{\text{tr}(\boldsymbol{\Sigma}_a\boldsymbol{\Sigma}_b)}{\sqrt{\text{tr}\,\boldsymbol{\Sigma}_a^2\,\text{tr}\,\boldsymbol{\Sigma}_b^2}}. \quad (15)$$

We first consider $\boldsymbol{\Sigma}_a = \boldsymbol{\Sigma}_b = \mathbf{I}_d$ in which case both populations perfectly align with a CKA $= 1$. As indicated by Equation (12), however, despite the perfect alignment of populations,

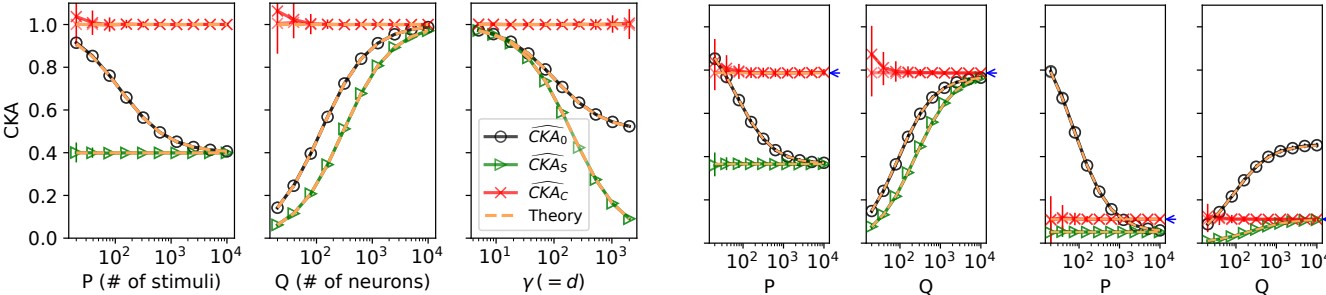

(a) Left: Varying $P$, with $Q = 200$ and $d = 300$. Middle: Varying $Q$, with $P = 200$ and $d = 300$. Right: Varying the intrinsic dimensionality $\gamma$, which equals $d$ in this setup, with $P = Q = 200$. The true CKA is 1.

(b) CKA between well-aligned representations with power-law spectra. Blue arrow: true CKA.

(c) CKA between misaligned representations with power-law spectra. Blue arrow: true CKA.

Figure 2: CKA estimators on the linear CKA example. The blue horizontal line is the true CKA. The vertical error bar indicates the range of the first and third quartiles of the data (50% of the data). The darker lines are $\widehat{\text{CKA}}$'s and the lighter lines are $\widehat{\text{CKA}^N}$'s where $N = 500$. The yellow dotted lines (labeled as 'Theory') are the theoretical predictions: for $\widehat{\text{CKA}}_C$, this is simply 1; for $\widehat{\text{CKA}}_S$, we use Equation (12); for $\widehat{\text{CKA}}_0$, we use Equation (S16).

the stimulus-corrected estimator $\widehat{\text{CKA}}_S$ can be arbitrarily small depending on the number of neurons sampled $Q$, and the dimensionality $\gamma$, which is exactly $d$ in this example. We numerically confirm these findings in Figure 2a and find that both estimators become highly sensitive when a limited number of neurons are observed. On the other hand, our estimator is able to recover the true CKA even at small neuron samples.

We also simulate biologically relevant representations, whose spectra follow a power-law: $\lambda_n = n^{-r}$ Stringer et al. (2019). In Figures 2b and 2c, we set the powers $r$ of the two representations to be $0.5$ and $0.9$, with $d = 1000$. In Figure 2b, we simply let $\mathbf{\Sigma}_a$ and $\mathbf{\Sigma}_b$ be diagonal matrices with these eigenvalues on the diagonals. In Figure 2c, we reverse the order of the diagonal entries of $\mathbf{\Sigma}_b$ to make them misaligned.

In these examples, we observe that $\widehat{\text{CKA}^N}$'s and $\widehat{\text{CKA}}$'s almost completely overlap for all three estimators (Figure 2 lighter vs. darker lines), except when $Q$ is very small, $\widehat{\text{CKA}^N}_C$ and $\widehat{\text{CKA}}_C$ diverge a bit. This means that the bias correction for $\mathcal{H}$ contributes more to the bias correction of CKA than correcting the bias from the non-linear operations on the $\mathcal{H}$ estimates. The next section shows that this pattern is observed in neural data as well.

### Practical applications in neuroscience

In most of the brain recordings, the observed neurons are samples of a much larger population. Here, we use electrophysiological data of the three key cortical regions for visual processing, V1, V4, and IT, in the order of the processing cascade. It has been shown that the neurons in V1 are simple filters, but the later regions V4 and IT are sensitive to semantic information (Hubel & Wiesel, 1962; Majaj et al., 2015; Hung et al., 2005; Cadena et al., 2024). Papale et al. (2025) presented 20,000 natural images of 1,854 object categories from the THINGS dataset (Hebart et al., 2019) to two monkeys and recorded neural responses with multiple electrode arrays over

V1, V4, and IT, totaling 1046 electrodes in one monkey and 960 in the other. The measurement value is the average spiking voltage levels of neurons adjacent to a given electrode, averaged over a small time window immediately following an image presentation. We view the electrodes and images as rows and columns, respectively.

### Brain-to-Brain alignment

Here, use CKA to benchmark the similarity of a given brain region across individual animals (see Figure 1a). Such brain-to-brain comparisons provide an essential reference for evaluating model-to-brain alignments. In our analysis, we first compare two disjoint sets of electrodes from the same brain region within a single animal. The ground truth in this within-region comparison is a perfect alignment (CKA = 1) on average. However, as shown in Figure 3a, both the naive estimator $\widehat{\text{CKA}}_0$ and the stimulus-corrected estimator $\widehat{\text{CKA}}_S$ significantly underestimate the true similarity, with their estimates strongly dependent on the number of neurons sampled. In contrast, our proposed estimator $\widehat{\text{CKA}}_C$ closely approximates the true value even with very limited neuron sampling, thereby highlighting the risk that conventional CKA estimates may fail to detect even perfect alignment when undersampling is present.

We further extend the analysis by measuring the similarity of each brain region across animals. As depicted in Figure 3b, $\widehat{\text{CKA}}_C$ yields reliable estimates over a range of neuron sampling sizes, whereas the conventional estimators remain highly sensitive to the sample size. Our results show that V1 representations are relatively conserved across individuals (with CKA around 0.6), IT representations exhibit lower similarity (around 0.3), and V4 lies in between (approximately 0.45). The similarity decreases over the layers of visual processing.

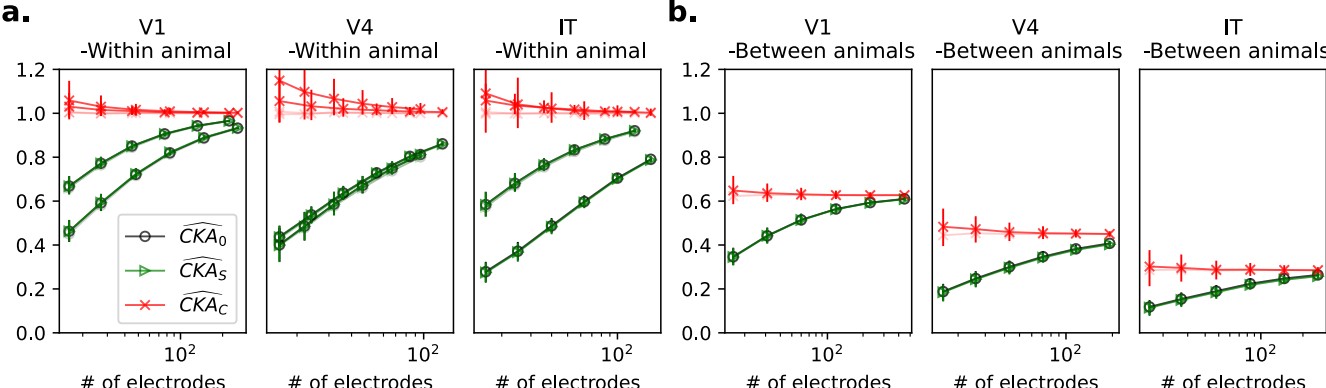

Figure 3: **a)** Each plot shows CKA values between disjoint sets of electrodes sampled from one brain region within one individual animal. The number of sampled electrodes ($Q$) is varied. One line for each animal. True CKA is 1. **b)** Each plot shows CKA values between animals for one brain region. $\widehat{CKA_0}$ and $\widehat{CKA_S}$ are similar, since the number of stimuli is large: $P = 2000$. The darker lines are $\widehat{CKA}$'s and the lighter lines are $\widehat{CKA^N}$'s where $N = 1000$.

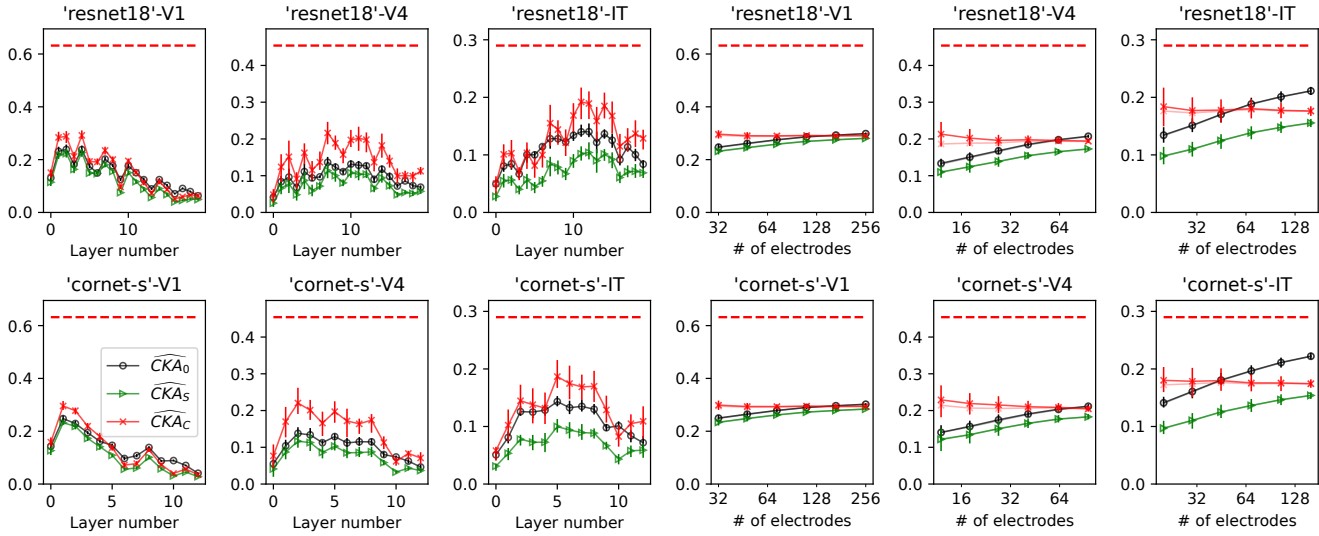

Figure 4: Comparison of the brain regions (V1,V4, and IT) and models (ResNet18 and CorNet-s) using CKA estimators. The left three columns show CKAs between each brain region and all model layers. The number of electrodes ($Q$) is downsampled to $1/16$ of all available electrodes in the dataset for each brain region. The right three columns show CKA between a brain region and the layer that is the most similar to the region. The number of electrodes ($Q$) varies from the factor of $1/16$ to $1$ (of all available electrodes) along the x-axis for each region. $P = 1,000$ stimuli are used in all plots. The horizontal dotted line is the brain-to-brain $\widehat{CKA_C}$ across animals for each brain region from Figure 3b. The darker lines are $\widehat{CKA}$'s and the lighter lines are $\widehat{CKA^N}$'s, where $N = 200$. The result for another monkey is shown in Appendix C.1.

## Brain-to-Model alignment

The alignment between brain representations and artificial neural network layers is a topic of considerable interest. It has been repeatedly observed in the literature that the early visual regions, e.g., V1, have representations that are strikingly similar to the early layers of CNNs, and the later regions, e.g., IT, are aligned with the deeper layers (Yamins et al., 2014; Schrimpf et al., 2018; Nonaka et al., 2021). As depicted in Figure 1b, here we use the CKA estimators to compare the represen-

tations between brain regions (V1, V4, and IT) and layers of convolutional neural networks (CNNs), namely ResNet18 and CorNet-s (He et al., 2016; Kubilius et al., 2018). We then see how these CKA estimates compare against the brain-to-brain $\widehat{CKA_C}$ from the previous section.

We make the same observation of the V1-early layer alignment and IT-late layer alignment from all of the estimators. In the absolute scale, the V1-early layer alignment is generally higher, but the IT-late layer alignment is generally lower

(Figure 4 left three plots). However, $\widehat{\mathrm{CKA}}_C$ estimates that the alignment between animals also decreases by a similar factor. This indicates that in the relative scale, the model-to-brain alignments for all three regions are similarly close to the animal-to-animal alignments, an insight that cannot be reliably found with the other estimators.

We then test the sensitivity of the estimators to the number of electrodes. In Figure 4 (right three plots), we pick the best alignment layer for each brain region and see how this alignment value changes with the number of electrodes. We observe that all estimators are similar for comparison with V1, meaning that there is only a small bias in $\widehat{\mathrm{CKA}}_0$ and $\widehat{\mathrm{CKA}}_S$. However, for the comparison with V4 and IT, the gap between the estimators is generally large, meaning we need our estimator $\widehat{\mathrm{CKA}}_C$ for a reliable measurement. The $\widehat{\mathrm{CKA}}_0$ and $\widehat{\mathrm{CKA}}_S$ values are not converged even when all available electrodes are used.

### Object disentanglement

Beyond inter-region comparisons, our estimator enables a novel application of CKA: quantifying the disentanglement of semantic information across object categories in the brain. Similar to the representation disentanglement observed in deeper layers of CNNs, the IT cortical region is believed to encode high-level semantic features that distinguish between objects (Yamins et al., 2014). In this analysis, we estimate the CKA between pairs of object categories within a single brain region, where the columns of the measurement matrix correspond to stimulus images and the rows correspond to the shared neurons (as illustrated in Figure 1b). A lower CKA in this context suggests greater semantic disentanglement.

We observe that the estimated CKA between object categories generally decreases over V1, V4, and IT, indicating a gradual semantic disentanglement over these regions (Figure 5). Interestingly, natural object pairs get separated faster than artificial object pairs. This pattern is more strongly observed in $\widehat{\mathrm{CKA}}_C$ than in $\widehat{\mathrm{CKA}}_S$. Interestingly, there are many cases where $\widehat{\mathrm{CKA}}_S$ indicates high disentanglement (small CKA) but ours $\widehat{\mathrm{CKA}}_C$ indicates low disentanglement (large CKA). This is observed in V1 for all pairwise comparisons: Our estimator consistently estimates CKA value near 1, suggesting V1 does not encode semantic information, whereas the heavily biased $\widehat{\mathrm{CKA}}_S$ spuriously suggests otherwise. Also, in IT, $\widehat{\mathrm{CKA}}_C$ is near 1 for artificial objects pairs, but the bias in $\widehat{\mathrm{CKA}}_S$ spuriously suggests the objects are disentangled.

### Trial-to-Trial Similarity

As our final application, we evaluate the trial-to-trial similarity of neural recordings from the same brain region on the same stimulus set as depicted in Figure 1c. We use the neural recordings from Papale et al. (2025) of visual cortical areas V1, V4, and IT over 30 trials on 100 images. This case differs from the previous analyses since a single population $\boldsymbol{\phi}^{(a)}$ is compared across two trials ($\boldsymbol{\phi}_1^{(a)}$ and $\boldsymbol{\phi}_2^{(a)}$). In this case, $\widehat{\mathrm{CKA}}_S$

becomes biased in its numerator as well as its denominator, which our estimator accounts for correctly.

In Figure 6, we calculate the CKA between all pair-wise single-trial measurements for each brain region and report the mean of each estimator. As before, our estimator yields consistent estimates across different neuron samplings $Q$, while the others remain highly biased. Furthermore, we observe that trial-to-trial similarities for regions V1 and V4 are significantly higher than those for region IT. This indicates that the trial-to-trial variability in IT is larger than in earlier visual regions.

### Discussion

Murphy et al. (2024) also highlights the sensitivity of the naive CKA estimator to the number of features, with a focus on the ratio of the number of inputs and features. They motivate the problem by showing that the naive CKA (Equation (1)) between two independent random matrices $X$ and $Y$ (their entries are i.i.d. standard normal) takes a non-zero value which depends on the ratio of the number of stimuli and neurons:

$$\frac{1}{\sqrt{\left(1 + \frac{P}{Q_a}\right)\left(1 + \frac{P}{Q_b}\right)}} \tag{16}$$

in the limit of large $P$, $Q_a$, and $Q_b$. They observe that $\widehat{\mathrm{CKA}}_S$ takes the value 0 in the same setup, resolving this issue of having spurious similarities between independent random matrices. While this observation is correct, Murphy et al. (2024) interprets this result as $\widehat{\mathrm{CKA}}_S$ resolving the issue of the naive CKA being sensitive to $Q_a$ and $Q_b$ (denoted by $P_1$ and $P_2$ in their paper), since $\widehat{\mathrm{CKA}}_S$ value is 0 regardless of $Q_a$ and $Q_b$. Extrapolating this interpretation, they apply $\widehat{\mathrm{CKA}}_S$ to real neural data in an attempt to resolve the issue of CKA sensitivity to the number of neurons.

However, here we explain that this interpretation is only valid when the true CKA is 0, and therefore cannot be generalized to other cases. $\widehat{\mathrm{CKA}}_S$ returns 0 in the random matrix setup, not because it corrects neuron sampling, but because it corrects the stimulus sampling. We can see that by taking the number of stimuli to infinity ($P \to \infty$) in Equation (16), which makes the CKA estimate approach 0. In practical scenarios, where the measurement matrices are not random, $\widehat{\mathrm{CKA}}_S$ is still sensitive to the number of features as we have shown in Equation (12). $\widehat{\mathrm{CKA}}_C$ presented in this paper resolves this problem.

Finally, our estimator for the $\mathcal{H}$-value can be used for debiasing other HSIC-based CKA measures, such as angular-CKA introduced in Williams et al. (2021) and Representational Similarity Analysis (RSA) (Kriegeskorte et al., 2008), since these measures also ignore the bias coming from finite neuron sampling.

### Conclusion

We have addressed a key limitation in applying CKA to scenarios where only a subset of features—such as neurons or model units—is observable. While CKA is widely used to compare

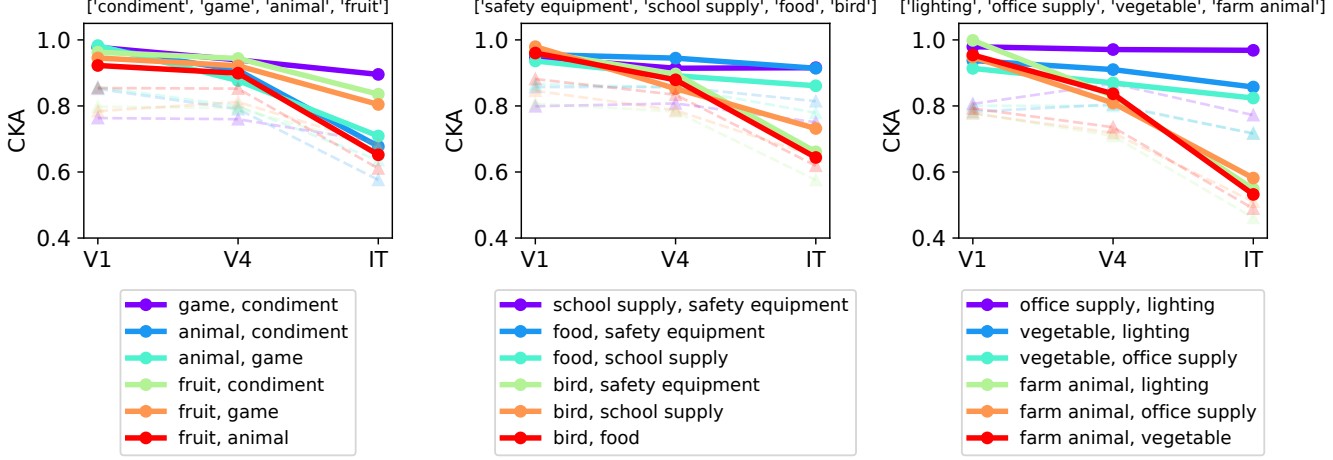

Figure 5: Quantifying semantic disentanglement in the brain with CKA. Solid line represents $\widehat{\text{CKA}}_C$, whereas dotted line represents $\widehat{\text{CKA}}_S$. Three separate groups of object images were prepared, and all pairwise comparisons were performed in each group. The result for another monkey is shown in the Appendix C.2.

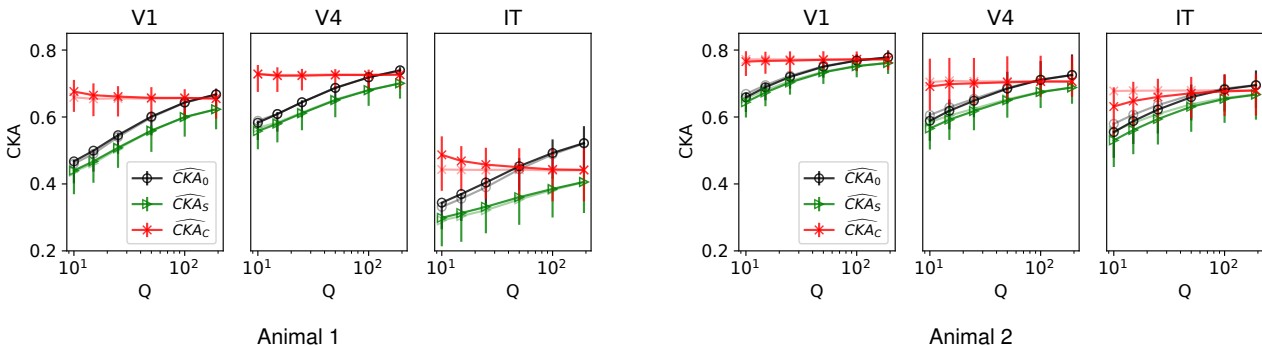

Figure 6: Trial-to-trial CKA estimates of brain regions V1, V4 and IT as a function of neuron sample size $Q$. Measurements on two animals are shown separately in the left plot and right plot. The darker lines are $\widehat{\text{CKA}}$'s and the lighter lines are $\widehat{\text{CKA}}^N$'s.

representations in machine learning and neuroscience, existing estimators systematically underestimate similarity when features are undersampled, a common issue in neural data collection or reduced-dimension network analyses.

Theoretically, we showed how the stimulus-corrected $\mathcal{H}$ estimator still fails to yield unbiased CKA under partial column sampling. We derived a bias-corrected estimator that handles both input and feature sampling, more accurately recovering true representation similarity. Empirically, our method performed well on both synthetic data and electrophysiological recordings from the ventral visual stream (V1, V4, and IT). Conventional approaches underestimated alignment, particularly for high-dimensional or semantically complex representations, whereas our estimator revealed consistent neural similarity patterns and clearer evidence of object-category disentanglement.

## Acknowledgments

Partial support for this work was provided by the Simons Foundation. DDL was also supported by the Institute of Information & Communications Technology Planning & Evaluation (IITP) grant funded by the Korean Government (MSIT) (No. RS-2024-00457882, National AI Research Lab Project). S.C. was partially supported by a Sloan Research Fellowship, a Klingenstein-Simons Award, and the Samsung Advanced Institute of Technology project, "Next Generation Deep Learning: From Pattern Recognition to AI." All experiments were performed using the Flatiron Institute's high-performance computing cluster.

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

# Supplementary Information

## A  Full derivation of the $\mathcal{H}$ estimators

### A.1  Expression of $\mathcal{H}$ in terms of the uncentered kernel

Recall that the kernel is defined as $k(x,y) = \langle \phi(x), \phi(y) \rangle$ and the centered kernel is $k'(x,y) = \langle \phi(x) - \mathbb{E}_x \left[ \phi(x) \right], \phi(y) - \mathbb{E}_x \left[ \phi(x) \right] \rangle$. Then $\mathcal{H}$ is defined as

$$\mathcal{H}(k^{(a)}, k^{(b)}) = \mathbb{E}_{x,y} \left[ k'^{(a)}(x,y) k'^{(b)}(x,y) \right]$$

Note that $k'$ can be written in terms of $k$:

$$k'(x,y) = \left\langle \phi(x) - \mathbb{E}_x \left[ \phi(x) \right], \phi(y) - \mathbb{E}_x \left[ \phi(x) \right] \right\rangle$$

$$= k(x,y) - \mathbb{E}_z \left[ k(x,z) \right] - \mathbb{E}_z \left[ k(z,y) \right] + \mathbb{E}_{z,w} \left[ k(z,w) \right].$$

Plugging this into our definition of $\mathcal{H}$, we arrive at the expression of $\mathcal{H}$ explicitly in terms of the original uncentered kernels, $k^{(a)}$ and $k^{(b)}$:

$$\mathcal{H}(k^{(a)}, k^{(b)}) = \mathbb{E}_{x,y} \left[ k^{(a)}(x,y) k^{(b)}(x,y) \right] - 2\mathbb{E}_{x,y,z} \left[ k^{(a)}(x,y) k^{(b)}(x,z) \right] + \mathbb{E}_{x,y} \left[ k^{(a)}(x,y) \right] \mathbb{E}_{x,y} \left[ k^{(b)}(x,y) \right]$$

We want to derive an estimator where each term has an expected value that is equal to an individual term above.

### A.2  Identifying the source of bias in the naive estimator

We begin our analysis by considering the naive $\mathcal{H}$ estimator:

$$\widehat{\mathcal{H}_0}(\Phi^{(a)}, \Phi^{(b)}) = \frac{1}{P^2} \text{tr} \left( H K^{(a)} H K^{(b)} \right)$$

where $H = I - \frac{1}{P} 11^\top$ and $K^{(a)} = \frac{1}{Q} \Phi^{(a)} \Phi^{(a)\top}$. When we expand this, we get

$$\widehat{\mathcal{H}_0}(\Phi^{(a)}, \Phi^{(b)}) = \frac{1}{P^2} \sum_{ij=1}^{P} K_{ij}^{(a)} K_{ij}^{(b)} - \frac{2}{P^3} \sum_{ijl=1}^{P} K_{ij}^{(a)} K_{jl}^{(b)} + \frac{1}{P^4} \sum_{ijlm=1}^{P} K_{ij}^{(a)} K_{lm}^{(b)}.$$

Note that $K^{(a)}$ and $K^{(b)}$ are dependent on the stimuli $\{x_i\}_{i=1}^{P}$ such that, for example $K_{ij}^{(a)} = k^{(a)}(x_i, x_j)$. Now, let us take the average of $\widehat{\mathcal{H}_0}$ over all possible stimuli sets $\{x_i\}_{i=1}^{P}$ sampled from $\mathcal{X}$. We can compute the expected value separately for each term and add them together for the final result. Let us consider the first term:

$$\mathbb{E}_{\{x_i\}_{i=1}^{P}} \left[ \frac{1}{P^2} \sum_{ij=1}^{P} K_{ij}^{(a)} K_{ij}^{(b)} \right] = \frac{1}{P^2} \mathbb{E}_{\{x_i\}_{i=1}^{P}} \left[ \sum_{ij=1}^{P} k^{(a)}(x_i, x_j) k^{(b)}(x_i, x_j) \right]$$

$$= \frac{1}{P^2} \sum_{i \neq j}^{P} \mathbb{E}_{x,x'} \left[ k^{(a)}(x,x') k^{(b)}(x,x') \right] + \frac{1}{P^2} \sum_{i}^{P} \mathbb{E}_x \left[ k^{(a)}(x,x) k^{(b)}(x,x) \right]$$

$$= \left( \frac{P-1}{P} \right) \mathbb{E}_{x,x'} \left[ k^{(a)}(x,x') k^{(b)}(x,x') \right] + \frac{1}{P} \mathbb{E}_x \left[ k^{(a)}(x,x) k^{(b)}(x,x) \right]$$

$$= \mathbb{E}_{x,x'} \left[ k^{(a)}(x,x') k^{(b)}(x,x') \right] + \frac{1}{P} \left( \mathbb{E}_x \left[ k^{(a)}(x,x) k^{(b)}(x,x) \right] - \mathbb{E}_{x,x'} \left[ k^{(a)}(x,x') k^{(b)}(x,x') \right] \right)$$

Notice that in the second equality, we separate the sum into two parts, based on whether the indices overlap or not: $i = j$ vs $i \neq j$. This is essential since the expected values are different for these two cases. Note that in the second term here is $O(1/P)$, contributing as bias. We would not have gotten this bias if the term $\frac{1}{P^2} \sum_{ij=1}^{P} K_{ij}^{(a)} K_{ij}^{(b)}$ was instead defined as $\frac{1}{P(P-1)} \sum_{i \neq j}^{P} K_{ij}^{(a)} K_{ij}^{(b)}$. In that case, the expected value does not have a bias:

$$\mathbb{E}_{\{x_i\}_{i=1}^{P}} \left[ \frac{1}{P(P-1)} \sum_{i \neq j}^{P} K_{ij}^{(a)} K_{ij}^{(b)} \right] = \mathbb{E}_{x,x'} \left[ k^{(a)}(x,x') k^{(b)}(x,x') \right].$$

Note that $\frac{1}{P(P-1)}$ is introduced as a scaling factor since $P(P-1)$ is the number of summand in $\sum_{i \neq j}^{P}$.

## A.3 Derivation of the stimulus-corrected $\mathcal{H}$ estimator

Applying the same logic to the rest of the terms in $\widehat{\mathcal{H}_0}(\Phi^{(a)}, \Phi^{(b)})$, we arrive at an estimator that removes the bias from input sampling:

$$\widehat{\mathcal{H}_S}(\Phi^{(a)}, \Phi^{(b)}) = \frac{1}{P(P-1)} \sum_{i \neq j}^{P} K_{ij}^{(a)} K_{ij}^{(b)} - \frac{2}{P(P-1)(P-2)} \sum_{i \neq j \neq l}^{P} K_{ij}^{(a)} K_{jl}^{(b)} + \frac{1}{P(P-1)(P-2)(P-3)} \sum_{i \neq j \neq l \neq m}^{P} K_{ij}^{(a)} K_{lm}^{(b)}.$$

This is equivalent to the estimator by Song et al. Note that these sums over disjoint indices, e.g. $\sum_{i \neq j \neq l}^{P}$, are practically difficult to compute, so they are decomposed into a linear combination of regular sums. Denoting $K_{ij}^{(a)} K_{lm}^{(b)}$ as $v_{ijlm}$, the first term can be rewritten as:

$$\sum_{i \neq j}^{P} K_{ij}^{(a)} K_{ij}^{(b)} = \sum_{ij} v_{ijij} - \sum_i v_{iiii},$$

whereas the second term can be:

$$\sum_{i \neq j \neq l}^{P} K_{ij}^{(a)} K_{jl}^{(b)} = \sum_{ijl} v_{ijjl} - \sum_{ij} v_{iiij} - \sum_{ij} v_{ijjj} - \sum_{ij} v_{ijij} + \sum_i 2 v_{iiii},$$

and the third term:

$$\sum_{i \neq j \neq l \neq m}^{P} K_{ij}^{(a)} K_{lm}^{(b)} = \sum_{ijlm} v_{ijlm} - \sum_{ijl} \left( v_{iijl} + v_{jlii} + 4 v_{ijjl} \right) + \sum_{ij} \left( v_{iijj} + 4 v_{iiij} + 4 v_{ijjj} + 2 v_{ijij} \right) - \sum_i 6 v_{iiii}$$

Replacing the terms in $\widehat{\mathcal{H}_S}(\Phi^{(a)}, \Phi^{(b)})$ with these new notations, we arrive at the following expression:

$$\widehat{\mathcal{H}_S}(\Phi^{(a)}, \Phi^{(b)}) = \frac{1}{P(P-3)} \times$$
$$\left( \left( \sum_{ij} v_{ijij} - \sum_i v_{iiii} \right) - \frac{2}{(P-2)} \left( \sum_{ijl} v_{ijjl} - \sum_{ij} v_{iiij} - \sum_{ij} v_{ijjj} + \sum_i v_{iiii} \right) + \frac{1}{(P-1)(P-2)} \left( \sum_{ijlm} v_{ijlm} - \sum_{ijl} v_{iijl} - \sum_{ijl} v_{jlii} + \sum_{ij} v_{iijj} \right) \right)$$
(S1)

$$\widehat{\mathcal{H}_S}(\Phi^{(a)}, \Phi^{(b)}) = \frac{1}{P^3(P-3)} \times$$
$$\sum_{ijlm} \left( \left( v_{ijij} - \frac{v_{iiii}}{P} \right) - \frac{2P}{P-2} \left( v_{ijjl} - \frac{v_{iiij}}{P} - \frac{v_{ijjj}}{P} + \frac{v_{iiii}}{P^2} \right) + \frac{P^2}{(P-1)(P-2)} \left( v_{ijlm} - \frac{v_{iijl}}{P} - \frac{v_{jlii}}{P} + \frac{v_{iijj}}{P^2} \right) \right) \quad \text{(S2)}$$

Suppose $K'$ is a version of $K$ whose diagonal elements are 0, and $v'_{ijlm} := K'^{(a)}_{ij} K'^{(b)}_{lm}$. Then, the above expression simplifies to

$$\widehat{\mathcal{H}_S}(\Phi^{(a)}, \Phi^{(b)}) = \frac{1}{P(P-3)} \left( \sum_{ij} v'_{ijij} - \frac{2}{(P-2)} \sum_{ijl} v'_{ijjl} + \frac{1}{(P-1)(P-2)} \sum_{ijlm} v'_{ijlm} \right)$$

$$= \frac{1}{P(P-3)} \left( \text{tr} \left( K'^{(a)} K'^{(b)} \right) - \frac{2}{(P-2)} 1^\top K'^{(a)} K'^{(b)} 1 + \frac{1}{(P-1)(P-2)} 1^\top K'^{(a)} 1 1^\top K'^{(b)} 1 \right)$$

which is the exact expression found in Song et al. (2012).

## A.4 Derivation of the stimulus-neuron-corrected $\mathcal{H}$ estimator

Our estimator assumes that the features are also sampled, and the features are correlated or identical in (a) and (b). Let us consider a single term $v_{ijlm}$ from Equation (S2) (redefined here as $K_{ij}^{(a)}K_{lm}^{(a)}$ reflecting $a=b$) and see how the sampling of features contribute to the bias.

$$v_{ijlm} = K_{ij}^{(a)}K_{lm}^{(a)} = \frac{1}{Q^2}\sum_{\alpha\beta}\Phi_{i\alpha}^{(a)}\Phi_{j\alpha}^{(a)}\Phi_{l\beta}^{(a)}\Phi_{m\beta}^{(a)}$$

From here on, we will drop the superscript $(a)$. If we were to take the expected value of $v_{ijlm}$ over the feature sampling, we need to define how exactly the features are sampled. To this end, we assume that each entry $\Phi_{i\alpha}$ is determined by

$$\Phi_{i\alpha} = \phi(x_i, w_\alpha) \tag{S3}$$

where $x_i$ is some random variable representing an $i$th stimulus, and $w_\alpha$ is some (abstract) latent random variable representing an $\alpha$th neuron. Both $x_i$ and $w_\alpha$ are assumed to be sampled randomly from their respective distributions. More details on these assumptions are provided in Appendix B. Note that

$$k(x_i, x_j) = \mathbb{E}_w\left[\phi(x_i, w)\phi(x_j, w)\right] \tag{S4}$$

We want to find an expression for some estimator $c'_{ijlm}$, an alternative to $v_{ijlm}$, such that when we average it over the feature samples (i.e. $\{w_\alpha\}_{\alpha=1}^Q$), we get

$$\mathbb{E}_{\{w_\alpha\}_{\alpha=1}^Q}\left[c'_{ijlm}\right] = k(x_i, x_j)k(x_l, x_m). \tag{S5}$$

It is important to note that $v_{ijlm} \equiv K_{ij}K_{lm}$ is not $k(x_i, x_j)k(x_l, x_m)$ on average. It is, in fact, a biased estimate of $k(x_i, x_j)k(x_l, x_m)$:

$$\mathbb{E}_{\{w_\alpha\}_{\alpha=1}^Q}\left[v_{ijlm}\right] = \mathbb{E}_{\{w_\alpha\}_{\alpha=1}^Q}\left[\frac{1}{Q^2}\sum_{\alpha\beta}\phi(x_i, w_\alpha)\phi(x_j, w_\alpha)\phi(x_l, w_\beta)\phi(x_m, w_\beta)\right] \tag{S6}$$

$$= \frac{1}{Q^2}\mathbb{E}_{\{w_\alpha\}_{\alpha=1}^Q}\left[\sum_{\alpha\neq\beta}\phi(x_i, w_\alpha)\phi(x_j, w_\alpha)\phi(x_l, w_\beta)\phi(x_m, w_\beta) + \sum_{\alpha=1}^Q\phi(x_i, w_\alpha)\phi(x_j, w_\alpha)\phi(x_l, w_\alpha)\phi(x_m, w_\alpha)\right] \tag{S7}$$

$$= \frac{Q-1}{Q}\mathbb{E}_{w,w'}\left[\phi(x_i, w)\phi(x_j, w)\phi(x_l, w')\phi(x_m, w')\right] + \frac{1}{Q}\mathbb{E}_w\left[\phi(x_i, w)\phi(x_j, w)\phi(x_l, w)\phi(x_m, w)\right] \tag{S8}$$

$$= \frac{Q-1}{Q}k(x_i, x_j)k(x_l, x_m) + \frac{1}{Q}\mathbb{E}_w\left[\phi(x_i, w)\phi(x_j, w)\phi(x_l, w)\phi(x_m, w)\right] \tag{S9}$$

From above, it is clear that the bias comes from the summation where $\alpha = \beta$, i.e. $\sum_{\alpha=1}^Q$ (the second terms in Equations (S7) to (S9)), whereas the summation over $\alpha \neq \beta$ recovers the quantity of interest $k(x_i, x_j)k(x_l, x_m)$. Therefore, the following should be an unbiased estimator of $k(x_i, x_j)k(x_l, x_m)$:

$$c'_{ijlm} = \frac{1}{Q(Q-1)}\sum_{\alpha\neq\beta}\Phi_{i\alpha}^{(a)}\Phi_{j\alpha}^{(a)}\Phi_{l\beta}^{(a)}\Phi_{m\beta}^{(a)} = \frac{1}{Q(Q-1)}\left(\sum_{\alpha\beta}\Phi_{i\alpha}^{(a)}\Phi_{j\alpha}^{(a)}\Phi_{l\beta}^{(a)}\Phi_{m\beta}^{(a)} - \sum_{\alpha}\Phi_{i\alpha}^{(a)}\Phi_{j\alpha}^{(a)}\Phi_{l\alpha}^{(a)}\Phi_{m\alpha}^{(a)}\right).$$

In the main text we define $c_{ijlm}$, which is simply $c'_{ijlm}$ without the $\frac{1}{Q(Q-1)}$ factor, i.e. $c_{ijlm} = Q(Q-1)c'_{ijlm}$. Therefore, finally, our estimator can be expressed as

$$\widehat{\mathcal{H}}_C(\Phi^{(a)}, \Phi^{(a)}) = \frac{1}{P^3(P-3)Q(Q-1)} \times$$

$$\sum_{ijlm}\left(\left(c_{ijij} - \frac{c_{iiii}}{P}\right) - \frac{2P}{P-2}\left(c_{ijjl} - \frac{c_{iiij}}{P} - \frac{c_{ijjj}}{P} + \frac{c_{iiii}}{P^2}\right) + \frac{P^2}{(P-1)(P-2)}\left(c_{ijlm} - \frac{c_{iijl}}{P} - \frac{c_{jlii}}{P} + \frac{c_{iijj}}{P^2}\right)\right) \tag{S10}$$

# B  Bias analysis

## B.1  Generative process framework

Here we formalize the problem setup by defining a general formulation of the process that generates the measurement matrices. Consider a pair of measurement matrices $\Phi_{(a)}$ and $\Phi_{(b)}$ of two systems $a$ and $b$. Let $x_i \in X$ be latent variables for the $i$th rows of $\Phi_{(a)}$ and $\Phi_{(b)}$, and $u_\alpha \in \mathcal{U}$ and $v_\alpha \in \mathcal{V}$ be column-latent variables for the $\alpha$th column of $\Phi_{(a)}$, and $\Phi_{(b)}$, respectively. Let $P$ be the number of row-latent variables sampled, and $Q_a$ and $Q_b$ be the numbers of column-latent variables sampled. Let $\phi_a : X \times \mathcal{U} \to \mathbb{R}$ be a map that defines the measurement value. Then we assume the entries of $\Phi_{(a)} \in \mathbb{R}^{P \times Q_a}$ and $\Phi_{(b)} \in \mathbb{R}^{P \times Q_b}$ are defined as

$$\Phi_{i\alpha}^{(a)} = \phi_a(x_i, u_\alpha), \text{and } \Phi_{i\beta}^{(b)} = \phi_b(x_i, v_\beta).$$

In some cases, a latent space might be fully observed in the measurement. For example, $N$ dimensional layer of neural network activation is given by a feature map $\psi_{(a)} : X \to \mathbb{R}^N$, where $X$ is the input space, and the column latent set $\mathcal{U}$ of cardinality $N$ is fully observed. Here, $\mathcal{U}$ would be a set of trained neural network weights. The uncentered kernel would be simply $k'(x, x') = \frac{1}{N} \psi(x) \psi(x')^\top$. If the latent variables cannot be fully observed, we assume there are probability measures over the latent spaces: $\rho_X$ is the probability measures over $X$ and, and similarly for $\rho_\mathcal{U}$ and $\rho_\mathcal{V}$. Assume $\phi_a$ is square integrable w.r.t. $\rho_X$ and $\rho_\mathcal{U}$, and similarly $\phi_b$ is also square integrable. The associated uncentered kernels are defined as $k_a'(x, x') = \int d\rho_\mathcal{U}(u)\, \phi_a(x, u)\phi_a(x', u)$ and $k_b'(y, y') = \int d\rho_\mathcal{V}(v)\, \phi_b(y, v)\phi_b(y', v)$. We may also define the associated kernel integral operator:

$$T_k f = \int d\rho_X(x)\, k'(\cdot, x) f(x). \tag{S11}$$

Later, the eigenvalues of the operator $T_k$ will be relevant.

We also define associated covariance kernels: $\tilde{k}_a'(u, u') = \int d\rho_X(x)\, \phi_a(x, u)\phi_a(x, u')$ and $\tilde{k}_b'(v, v') = \int d\rho_X(x)\, \phi_b(x, v)\phi_b(x, v')$.

## B.2  Biases in CKA estimators contributed by the biases in HSIC estimators

Here we derive the analytical expression of the biases of the CKA estimators. Here we assume that each feature is already centered, i.e. $\int d\rho_X(x)\, \phi_a(x, \cdot) = 0$, and $\int d\rho_X(x)\, \phi_b(x, \cdot) = 0$. We first compute the expected values of the HSIC estimators. Let $S_X := \{x_i\}_{i=1}^P$, $\mathcal{F}_\mathcal{U} := \{u_i\}_{i=1}^{Q_a}$, and $\mathcal{F}_\mathcal{V} := \{v_i\}_{i=1}^{Q_b}$ be the sets of latent variables sampled independently from $\rho_X$, $\rho_\mathcal{U}$, and $\rho_\mathcal{V}$ respectively. The naive HSIC estimator is then given by

$$\widehat{\mathcal{H}_0}(\phi_a, \phi_b, S_X, \mathcal{F}_\mathcal{U}, \mathcal{F}_\mathcal{V}) = \frac{1}{P^2 Q_a Q_b} \sum_{ij} \sum_\alpha \sum_\beta \phi_a(x_i, u_\alpha)\phi_a(x_j, u_\alpha)\phi_b(x_i, v_\beta)\phi_b(x_j, v_\beta)$$

if $u_\alpha$ and $v_\beta$ are independently sampled for all $\alpha$ and $\beta$ combinations, which corresponds to the numerator of CKA. If the column latent variables are identical, and $\phi_a = \phi_b$, this corresponds to the HSICs in the denominator of naive CKA:

$$\widehat{\mathcal{H}_0}(\phi_a, S_X, \mathcal{F}_\mathcal{U}) = \frac{1}{P^2 Q_a^2} \sum_{ij} \sum_{\alpha\beta} \phi_a(x_i, u_\alpha)\phi_a(x_j, u_\alpha)\phi_a(x_i, u_\beta)\phi_a(x_j, u_\beta),$$

$$\widehat{\mathcal{H}_0}(\phi_b, S_X, \mathcal{F}_\mathcal{V}) = \frac{1}{P^2 Q_b^2} \sum_{ij} \sum_{\alpha\beta} \phi_b(x_i, v_\alpha)\phi_b(x_j, v_\alpha)\phi_b(x_i, v_\beta)\phi_b(x_j, v_\beta)$$

Here, we assume the naive CKA is computed after $N$ trials, across which the empirical means of the HSIC are computed::

$$\widehat{\mathrm{CKA}}_0(\phi_a, \phi_b, \rho_X, \rho_\mathcal{U}, \rho_\mathcal{V}) = \frac{\sum_{l=1}^N \widehat{\mathcal{H}_0}(\phi_a, \phi_b, S_X^{(l)}, \mathcal{F}_\mathcal{U}^{(l)}, \mathcal{F}_\mathcal{V}^{(l)})}{\sqrt{\sum_{l=1}^N \widehat{\mathcal{H}_0}(\phi_a, S_X^{(l)}, \mathcal{F}_\mathcal{U}^{(l)}) \sum_{l=1}^N \widehat{\mathcal{H}_0}(\phi_b, S_X^{(l)}, \mathcal{F}_\mathcal{V}^{(l)})}}$$

We assume $N$ is large such that all three $\widehat{\mathcal{H}}$ empirical averages have small variance of order $O\left(\frac{1}{N}\right)$. In this limit, the following approximation is valid:

$$\left\langle \widehat{\mathrm{CKA}}_0(\phi_a, \phi_b, S_X, \mathcal{F}_\mathcal{U}, \mathcal{F}_\mathcal{V}) \right\rangle \approx \frac{\left\langle \widehat{\mathcal{H}_0}(\phi_a, \phi_b, S_X^{(l)}, \mathcal{F}_\mathcal{U}^{(l)}, \mathcal{F}_\mathcal{V}^{(l)}) \right\rangle}{\sqrt{\left\langle \widehat{\mathcal{H}_0}(\phi_a, S_X^{(l)}, \mathcal{F}_\mathcal{U}^{(l)}) \right\rangle \left\langle \widehat{\mathcal{H}_0}(\phi_b, S_X^{(l)}, \mathcal{F}_\mathcal{V}^{(l)}) \right\rangle}}.$$

This approximation is also valid when $P$, $Q_a$, and $Q_b$ are all large. We empirically observe that the expected value of $\widehat{\mathrm{CKA}}_0$ obtained via this approximation still accurately predict the expected values of $\widehat{\mathrm{CKA}}_0$ empirically computed even with $N = 1$ and

often even in addition to small $P$, $Q_a$, and $Q_b$. This allows us to understand the bias of $\widehat{\text{CKA}}_0$ contributed from the biases of $\widehat{\mathcal{H}_0}$, isolated from the bias contributed by taking products, inverse, and square root of the $\widehat{\mathcal{H}_0}$ estimates, which unnecessarily complicates the analysis.

First, let us take the expected value of the $\widehat{\mathcal{H}_0}$ in the numerator:

$$\left\langle \widehat{\mathcal{H}_0}(\phi_a, \phi_b, \mathcal{S}_X, \mathcal{F}_\mathcal{U}, \mathcal{F}_\mathcal{V}) \right\rangle = \frac{1}{P^2 Q_a Q_b} \sum_{ij} \sum_\alpha \sum_\beta \left\langle \phi_a(x_i, u_\alpha)\phi_a(x_j, u_\alpha)\phi_b(x_i, v_\beta)\phi_b(x_j, v_\beta) \right\rangle \tag{S12}$$

$$= \left\langle k_a(x,y)k_b(x,y) \right\rangle_{x,y} + \frac{1}{P}\left( \left\langle k_a(x,x)k_b(x,x) \right\rangle_x - \left\langle k_a(x,y)k_b(x,y) \right\rangle_{x,y} \right) \tag{S13}$$

$$= \left\langle k_a(x,y)k_b(x,y) \right\rangle_{x,y} \left( 1 + \frac{1}{P}\left( \zeta_{ab} - 1 \right) \right) \tag{S14}$$

where we have introduced a new variable $\zeta_{ab} := \frac{\langle k_a(x,x)k_b(x,x)\rangle_x}{\langle k_a(x,y)k_b(x,y)\rangle_{x,y}}$ that is sensitive to the alignment of the representation. Note that the expected value of the stimulus-corrected estimator $\widehat{\mathcal{H}_S}(\phi_a, \phi_b, \mathcal{S}_X, \mathcal{F}_\mathcal{U}, \mathcal{F}_\mathcal{V})$ can be obtained by taking $P \to \infty$ limit in Equation (S14), which simply $\langle k_a(x,y)k_b(x,y) \rangle$, which means that the numerator $\widehat{\mathcal{H}_S}$ is unbiased in this problem setup. However, in an alternative problem setup, such as the trial-to-trial CKA, the numerator $\widehat{\mathcal{H}_S}$ is still biased, since the neurons are identical across (a) and (b), i.e. $u_\alpha = v_\alpha$. The bias of the numerator $\widehat{\mathcal{H}_S}$ in this alternative problem setup is similar to the biases in the denominator $\widehat{\mathcal{H}_S}$'s of both problem setup.

Next, let us take the expected value of one of the $\widehat{\mathcal{H}_0}$'s in the denominator:

$$\left\langle \widehat{\mathcal{H}_0}(\phi_a, \mathcal{S}_X, \mathcal{F}_\mathcal{U}) \right\rangle = \frac{1}{P^2 Q_a^2} \sum_{ij} \sum_{\alpha\beta} \left\langle \phi_a(x_i, u_\alpha)\phi_a(x_j, u_\alpha)\phi_a(x_i, u_\beta)\phi_a(x_j, u_\beta) \right\rangle$$

$$= \left\langle k_a(x,y)^2 \right\rangle - \frac{1}{P}\left( \left\langle k_a(x,y)^2 \right\rangle - \left\langle k_a(x,x)^2 \right\rangle \right) - \frac{1}{Q_a}\left( \left\langle k_a(x,y)^2 \right\rangle - \left\langle \tilde{k}_a(w,w)^2 \right\rangle \right) +$$
$$\frac{1}{PQ_a}\left( \left\langle k_a(x,y)^2 \right\rangle - \left\langle k_a(x,x)^2 \right\rangle - \left\langle \tilde{k}_a(w,w)^2 \right\rangle + \left\langle \phi_a(x,w)^4 \right\rangle \right) \tag{S15}$$

$$= \left\langle k_a(x,y)^2 \right\rangle \left[ 1 + \frac{1}{P}\left( \frac{\gamma_a}{\psi_a} - 1 \right) + \frac{1}{Q_a}\left( \frac{\gamma_a}{\tilde{\psi}_a} - 1 \right) - \frac{1}{PQ_a}\left( \frac{\gamma_a}{\psi_a} + \frac{\gamma_a}{\tilde{\psi}_a} - \frac{\gamma_a}{\rho_a} - 1 \right) \right]$$

where we have introduced new variables $\gamma_a = \frac{\langle k_a(x,x)\rangle^2}{\langle k_a(x,y)^2\rangle}$, $\psi_a := \frac{\langle k_a(x,x)\rangle^2}{\langle k_a(x,x)^2\rangle}$, $\tilde{\psi}_a := \frac{\langle \tilde{k}_a(w,w)\rangle^2}{\langle \tilde{k}_a(w,w)^2\rangle}$, and $\rho_a = \frac{\langle \phi_a(x,w)^2\rangle^2}{\langle \phi_a(x,w)^4\rangle}$. Each of them is participation ratio (PR), i.e. effective/soft count, of some quantities. For discrete quantities, let us define PR as

$$\frac{\left( \sum_i a_i \right)^2}{\sum_i a_i^2}.$$

It is easy to see that, if $N$ number of $a_i$'s take value $1$ and the rest $0$, then PR is $N$, indicating it is a soft count of non-zero $a_i$. The continuous version is

$$\frac{\left( \int d\mu(t)\, f(t) \right)^2}{\int d\mu(t)\, f(t)^2},$$

which we call PR of $f$ w.r.t. to $\mu$. With these definitions, we can interpret $\gamma_a$ as the PR of the eigenvalues of $T_{k_a}$, i.e. intrinsic dimensionality of $T_k$, $\psi_a$ as the PR of $\int d\rho_\mathcal{U}(w)\,\phi_a(\cdot, w)^2$ w.r.t. $\rho_X$, $\tilde{\psi}_a$ as the PR of $\int d\rho_X(x)\,\phi_a(x, \cdot)^2$ w.r.t. $\rho_\mathcal{U}$, and $\rho_a$ as the PR of $\phi_a^2$ w.r.t. $\rho_X \otimes \rho_\mathcal{U}$. Suppose $\Phi_\infty^{(a)}$ is the measurement matrix in the limit of infinite stimulus and neuron samples, and $\mathbf{K}_\infty^{(a)}$ is the corresponding Gram matrix, equivalent to $T_{k_a}$. Then we can loosely say $\gamma_a$ is the intrinsic dimensionality of $\mathbf{K}_\infty$, $\psi_a$ is the effective number of rows of $\Phi_\infty^{(a)}$ with non-zero lengths, $\tilde{\psi}_a$ is the effective number of columns of $\Phi_\infty^{(a)}$ with non-zero lengths, and $\rho_a$ is the effective number of non-zero entries in $\Phi_\infty^{(a)}$.

Then, with the expected values of $\widehat{\mathcal{H}_0}$ derived above the expected value of $\widehat{\text{CKA}}_0$ can be approximated as

$$\left\langle \widehat{\mathrm{CKA}}_0(\phi_a, \phi_b, \mathcal{S}_X, \mathcal{F}_\mathcal{U}, \mathcal{F}_\mathcal{V}) \right\rangle \approx \frac{\left(1 + \frac{1}{P}\left(\zeta_{ab} - 1\right)\right) \mathrm{CKA}}{\sqrt{\left(1 + \frac{\frac{\gamma_a}{\psi_a} - 1}{P} + \frac{\frac{\gamma_a}{\psi_a} - 1}{Q_a} - \frac{\frac{\gamma_a}{\psi_a} + \frac{\gamma_a}{\psi_a} - \frac{\gamma_a}{\rho_a} - 1}{PQ_a}\right)\left(1 + \frac{\frac{\gamma_b}{\psi_b} - 1}{P} + \frac{\frac{\gamma_b}{\psi_b} - 1}{Q_b} - \frac{\frac{\gamma_b}{\psi_b} + \frac{\gamma_b}{\psi_b} - \frac{\gamma_b}{\rho_b} - 1}{PQ_b}\right)}} \tag{S16}$$

where $\mathrm{CKA} := \frac{\langle k_a(x,y) k_b(x,y) \rangle}{\sqrt{\langle k_a(x,y)^2 \rangle \langle k_b(x,y)^2 \rangle}}$ is the true CKA. Taking $P \to \infty$, we obtain the expected value of $\widehat{\mathrm{CKA}}_S(\phi_a, \phi_b, \mathcal{S}_X, \mathcal{F}_\mathcal{U}, \mathcal{F}_\mathcal{V})$ estimator:

$$\left\langle \widehat{\mathrm{CKA}}_S(\phi_a, \phi_b, \mathcal{S}_X, \mathcal{F}_\mathcal{U}, \mathcal{F}_\mathcal{V}) \right\rangle \approx \frac{\mathrm{CKA}}{\sqrt{\left(1 + \frac{\frac{\gamma_a}{\psi_a} - 1}{Q_a}\right)\left(1 + \frac{\frac{\gamma_b}{\psi_b} - 1}{Q_b}\right)}}$$

If $\int d\rho_X(x)\, \phi_a(x, \cdot)^2$ and $\int d\rho_X(x)\, \phi_b(x, \cdot)^2$ are constant, i.e. the norm of the activation is normalized for each neuron, then $\tilde\psi_a = \tilde\psi_b = 1$, which gives

$$\left\langle \widehat{\mathrm{CKA}}_S(\phi_a, \phi_b, \mathcal{S}_X, \mathcal{F}_\mathcal{U}, \mathcal{F}_\mathcal{V}) \right\rangle \approx \frac{\mathrm{CKA}}{\sqrt{\left(1 + \frac{\gamma_a - 1}{Q_a}\right)\left(1 + \frac{\gamma_b - 1}{Q_b}\right)}}$$

## B.3  Analyzing the overall $\widehat{\mathcal{H}}$-based CKA estimators

All CKA estimators based on $\widehat{\mathcal{H}}$, including ours, have bias coming from taking the product of $\widehat{\mathcal{H}}$'s that are correlated to each other, taking inverse, and taking the square root. As a simple analog of CKA estimator, consider the following problem. We have fixed quantities $X$, $A$, and $B$, and corresponding potentially biased estimators $x$, $a$, and $b$, respectively. Here, $x$, $a$, and $b$ are correlated. We wish to study the properties of the estimator

$$T = \frac{x}{\sqrt{ab}},$$

as an estimator for

$$\theta = \frac{X}{\sqrt{AB}}.$$

Here, the fixed quantities $X$, $A$, and $B$ correspond to the ground truth $\mathcal{H}$ values, and $x$, $a$, and $b$ correspond to the $\widehat{\mathcal{H}}$ estimators.

We denote

$$\delta_x = E[x] - X, \quad \delta_a = E[a] - A, \quad \delta_b = E[b] - B,$$

$$\sigma_x^2 = \mathrm{Var}(x), \quad \sigma_a^2 = \mathrm{Var}(a), \quad \sigma_b^2 = \mathrm{Var}(b),$$

$$\sigma_{xa} = \mathrm{Cov}(x,a), \quad \sigma_{xb} = \mathrm{Cov}(x,b), \quad \sigma_{ab} = \mathrm{Cov}(a,b).$$

We also define the deviations:

$$\Delta x = x - X, \quad \Delta a = a - A, \quad \Delta b = b - B,$$

with

$$E[\Delta x] = \delta_x, \quad E[\Delta a] = \delta_a, \quad E[\Delta b] = \delta_b.$$

We consider the function

$$f(x,a,b) = \frac{x}{\sqrt{ab}},$$

and expand it about the point $(X, A, B)$ to second order:

$$f(x,a,b) \approx f(X,A,B) + f_x \Delta x + f_a \Delta a + f_b \Delta b + \frac{1}{2}\left[f_{xx}(\Delta x)^2 + f_{aa}(\Delta a)^2 + f_{bb}(\Delta b)^2 + 2f_{xa}\Delta x \Delta a + 2f_{xb}\Delta x \Delta b + 2f_{ab}\Delta a \Delta b\right]$$

with all derivatives evaluated at $(X, A, B)$.

The zeroth-order term:

$$f(X,A,B) = \frac{X}{\sqrt{AB}}.$$

The first order derivatives:

$$f_x(X,A,B) = \frac{1}{\sqrt{AB}}, \quad f_a(X,A,B) = -\frac{X}{2A\sqrt{AB}}, \quad f_b(X,A,B) = -\frac{X}{2B\sqrt{AB}}.$$

Next, the second-order derivatives:

$$f_{xx}(X,A,B) = 0, \quad f_{xa}(X,A,B) = -\frac{1}{2A\sqrt{AB}}, \quad f_{xb}(X,A,B) = -\frac{1}{2B\sqrt{AB}},$$

$$f_{aa}(X,A,B) = \frac{3X}{4A^2\sqrt{AB}}, \quad f_{bb}(X,A,B) = \frac{3X}{4B^2\sqrt{AB}}, \quad f_{ab}(X,A,B) = \frac{X}{4AB\sqrt{AB}}.$$

Plugging them into our expansion, we get

$$\frac{x}{\sqrt{ab}} \approx \frac{X}{\sqrt{AB}}\left(1 + \frac{\Delta x}{X} - \frac{1}{2}\frac{\Delta a}{A} - \frac{1}{2}\frac{\Delta b}{B} + \frac{3}{8}\frac{(\Delta a)^2}{A^2} + \frac{3}{8}\frac{(\Delta b)^2}{B^2} - \frac{1}{2}\frac{\Delta x \Delta a}{XA} - \frac{1}{2}\frac{\Delta x \Delta b}{XB} + \frac{1}{4}\frac{\Delta a \Delta b}{AB}\right)$$

Now let us take the expected value of the above. We use

$$E[\Delta x] = \delta_x, \quad E[\Delta a] = \delta_a, \quad E[\Delta b] = \delta_b,$$

$$E[(\Delta x)^2] = \sigma_x^2 + \delta_x^2, \quad E[(\Delta a)^2] = \sigma_a^2 + \delta_a^2, \quad E[(\Delta b)^2] = \sigma_b^2 + \delta_b^2,$$

$$E[\Delta x \Delta a] = \sigma_{xa} + \delta_x \delta_a, \quad E[\Delta x \Delta b] = \sigma_{xb} + \delta_x \delta_b, \quad E[\Delta a \Delta b] = \sigma_{ab} + \delta_a \delta_b.$$

With these definitions, we get the following as the expected value up to the second order approximation:

$$E\left[\frac{x}{\sqrt{ab}}\right] \approx \frac{X}{\sqrt{AB}}\left(1 + \frac{\delta_x}{X} - \frac{1}{2}\frac{\delta_a}{A} - \frac{1}{2}\frac{\delta_b}{B} + \frac{3}{8}\frac{\sigma_a^2 + \delta_a^2}{A^2} + \frac{3}{8}\frac{\sigma_b^2 + \delta_b^2}{B^2} - \frac{1}{2}\frac{\sigma_{xa} + \delta_x \delta_a}{XA} - \frac{1}{2}\frac{\sigma_{xb} + \delta_x \delta_b}{XB} + \frac{1}{4}\frac{\sigma_{ab} + \delta_a \delta_b}{AB}\right).$$

Therefore, the bias is given by

$$E\left[\frac{x}{\sqrt{ab}}\right] - \frac{X}{\sqrt{AB}} \approx \frac{X}{\sqrt{AB}}\left(\frac{\delta_x}{X} - \frac{1}{2}\frac{\delta_a}{A} - \frac{1}{2}\frac{\delta_b}{B} + \frac{3}{8}\frac{\sigma_a^2 + \delta_a^2}{A^2} + \frac{3}{8}\frac{\sigma_b^2 + \delta_b^2}{B^2} - \frac{1}{2}\frac{\sigma_{xa} + \delta_x \delta_a}{XA} - \frac{1}{2}\frac{\sigma_{xb} + \delta_x \delta_b}{XB} + \frac{1}{4}\frac{\sigma_{ab} + \delta_a \delta_b}{AB}\right).$$

Now let us put this back into the perspective of the CKA estimation. In that context, $\delta_x$ is the bias of an $\mathcal{H}$-estimator for the numerator, whereas $\delta_a$ and $\delta_b$ are the bias of the $\mathcal{H}$-estimators in the numerator of CKA estimator. The variances $\sigma_x^2$, $\sigma_a^2$, and $\sigma_b^2$ are the variances of the corresponding $\mathcal{H}$-estimators. Again, $X, A, B$ corresponds to the true $\mathcal{H}$-values in the true CKA. In our estimator $\widehat{\text{CKA}}_C$, the $\mathcal{H}_C$ biases $\delta_x$, $\delta_a$, and $\delta_b$ are zero, substantially reducing the bias in $\widehat{\text{CKA}}_C$. However, in $\widehat{\text{CKA}}_0$ and $\widehat{\text{CKA}}_S$, their $\mathcal{H}$ estimators have non-zero biases $\delta_x, \delta_a, \delta_b > 0$ of order $O\left(\frac{1}{P} + \frac{1}{Q}\right)$ (except, for some setups, $\delta_x$ is zero in $\widehat{\text{CKA}}_S$), adding even more bias to the CKA estimations. Note that all $\mathcal{H}$ estimators have variance ($\sigma_x^2$, $\sigma_a^2$, $\sigma_b^2$, $\sigma_{xa}$, $\sigma_{xb}$, and $\sigma_{ab}$) of order $O\left(\frac{1}{P} + \frac{1}{Q}\right)$. Therefore, as long as CKA estimation is built based on the $\mathcal{H}$ estimators, the overall bias of CKA is of order $O\left(\frac{1}{P} + \frac{1}{Q}\right)$.

In a scenario where we have two systems (a) and (b) and we aim to compare the representations in (a) and (b) for a single input distribution. However, for each trial, one uses a unique set of inputs and observes a unique set of neurons independently drawn from distributions. Then, instead of taking the empirical average of $\widehat{\text{CKA}}$, one can take the empirical averages of $\widehat{\mathcal{H}}$ and use that average to compute CKA. If there are $N$ trials, these empirical averages of $\widehat{\mathcal{H}}$ have the variance of order $O\left(\frac{1}{N}\left(\frac{1}{P} + \frac{1}{Q}\right)\right)$. This situation corresponds to when $N$ number of labs use distinct individual animals to study V1 and they use distinct sets of natural images. Alternatively, if one aims to compute trial-to-trial CKA and there are $M$ trials, then by performing all pairwise comparisons, one can reduce the bias of CKA estimators significantly since $N = \binom{M}{2}$. In summary, having multiple trials allows our estimator $\widehat{\text{CKA}}_C$ to have bias of $O\left(\frac{1}{N}\left(\frac{1}{P} + \frac{1}{Q}\right)\right)$, while the other estimators $\widehat{\text{CKA}}_0$ and $\widehat{\text{CKA}}_S$ still has bias of $O\left(\frac{1}{P} + \frac{1}{Q}\right)$.

# C Additional results on neural data

In this section, we present the results on the second monkey subject. Overall, our observations are consistent across the two monkeys.

## C.1 Brain-to-model alignment

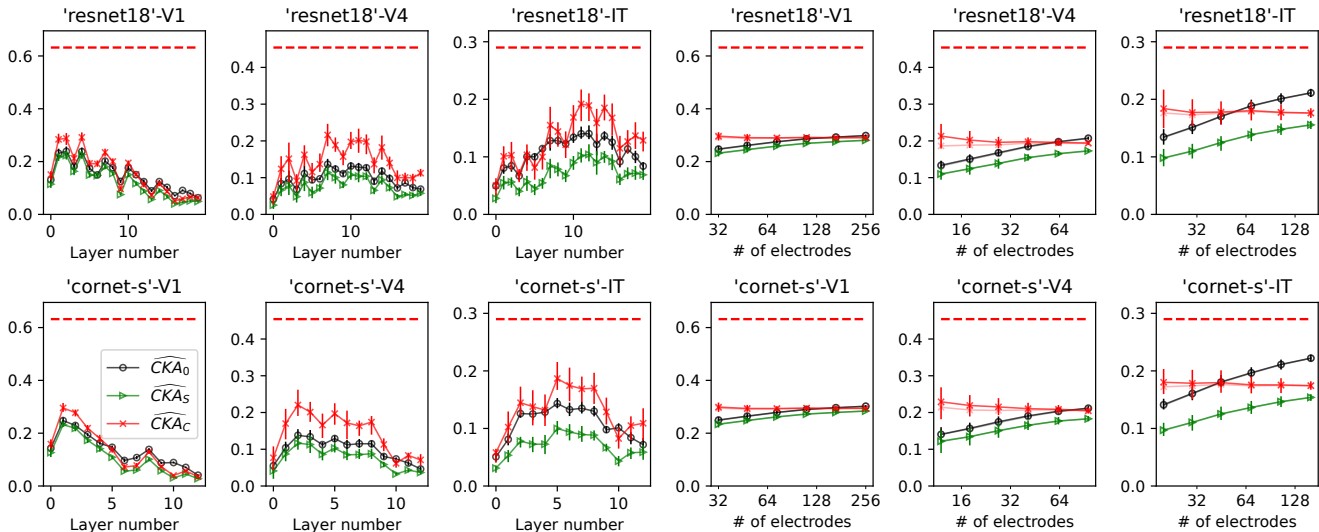

Figure S1: Comparison of the brain regions (V1,V4, and IT) and models (ResNet18 and CorNet-s) using CKA estimators. The left three columns show CKAs between each brain region and all model layers. The number of electrodes ($Q$) is downsampled to $1/16$ of all available electrodes in the dataset for each brain region. The right three columns show CKA between a brain region and the layer that is the most similar to the region. The number of electrodes ($Q$) varies from the factor of $1/16$ to 1 (of all available electrodes) along the x-axis for each region. $P = 1,000$ stimuli are used in all plots. The horizontal dotted line is the brain-to-brain $\widehat{\text{CKA}}_C$ across animals for each brain region from Figure 3b. The darker lines are $\widehat{\text{CKA}}$'s and the lighter lines are $\widehat{\text{CKA}}^N$'s.

The CKA estimates between CNNs and the second monkey is shown in Figure S1.

## C.2 Object disentanglement

The CKA estimates object disentanglement in the second monkey is shown in Figure S2.

# D Code availability

The code for the estimators and generating all figures is available in https://github.com/badooki/CKA/.

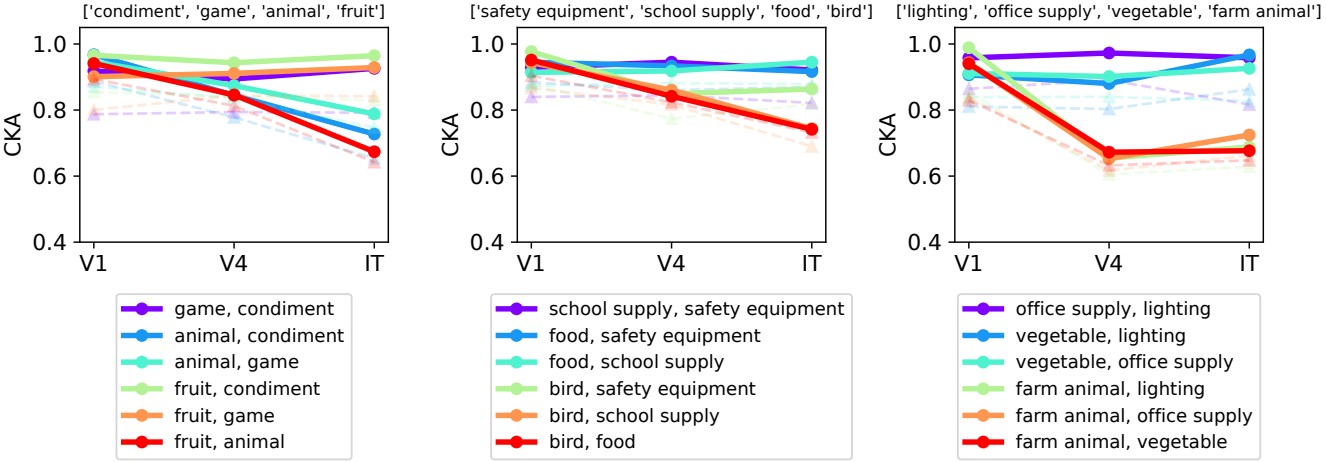

Figure S2: Quantifying semantic disentanglement in the brain with CKA. Solid line represents $\widehat{\mathrm{CKA}}_C$, whereas dotted line represents $\widehat{\mathrm{CKA}}_S$. Three separate groups of object images were prepared, and all pairwise comparisons were performed in each group.

