# OpenReview forum: "Estimating Neural Representation Alignment from Sparsely Sampled Inputs and Features"
_ccneuro.org/CCN/2025/Proceedings — CCN 2025 Proceedings asProceedingsPoster_

### Official Review · Reviewer_nows · 2025-03-31
**Nice work on an important methods question for comparing DNNs to the brain**

**Soundness:** 3
**Clarity:** 2

**Comments:**

Interest:  the authors address what I feel is important question for comparing neural network models to the brain (and indeed, comparing brains to each other): how do you correct biased mapping methods (and most relevant methods *are* biased) so that they are less susceptible to that bias at small numbers of neurons.   This problem is something we all suffer from and so addressing it is timely.

Clarity:  I didn't fully follow all the math in the appendix, especially around the justifications for the key correction formula 9.  I think this is probably mostly due to lack of effort on my part.  Maybe I would say that if the authors really wanted to be exceptional in this regard, some more intuition would need to be built up as to why their correction strategy is correct. All the key details are in the supplement, and treated kind of sparingly and formally.  I would have appreciate if the math could have been explained in a more intuitively satisfying way.   But the exposition for formulas 1-8 (which are like the standard material in the field) was fine, so I felt I basically understood the result of the paper.

Soundness:  I would say that this paper has strong soundness theoretically and adequate soundness empirically.   The mathematical analysis and derivation of their correction seems strong. (Though I would say I'm not 100% expert in that type of analysis myself.) Empirically the linear sanity check (Fig 2) was helpful.

In terms of the comparisons to neural data, the comparisons to neural data seemed clear and clean to me, and yielding reasonable results.  I especially found the comparisons of the estimation models on establishing "anatomical correspondence" useful (e.g. the ability of the corrected model to properly capture V4 peaks in Resnet-18 intermediate layers).   I would love to have seen this method on one more neural dataset (like, say, the NSD fMRI dataset), and computed for a few more models (Resnet 50, supervised ViT, etc).   One of the key results in the recent literature is that one needs to average over a number of models within a model class to see certain phenomena clearly (like anatomical correspondence), because there is a lot of model noise.  Is that something that would be cleaned up by this new work?  I would love to haveseen that, just by virtue of seeing their results on more models and datasets.

**Expertise:**

3

**Interest:**

2

---

> ### Author Rebuttal · Authors · 2025-04-15
>
> We thank the reviewer for carefully reading our manuscript and providing valuable feedback.
>
> We agree with the reviewer that the correction in formula 9 is not clearly explained. There exists a more direct and intuitive explanation of the derivation, which we will incorporate in the final version of our manuscript. Meanwhile, we tried to provide an intuitive explanation in line 206 of our manuscript, but we will make it more transparent in the revised version of our manuscript.
>
> We can readily perform the identical analysis on the fMRI dataset (stimuli: THINGS image dataset) and other model architectures, thanks to the THINGS initiative and the THINGSvision python package (by Martin Hebart’s group). We will provide the additional analyses in the final version of our appendix. Without the numerical experiment, we cannot claim that our CKA estimator is robust to different model instantiations of a single model class, since it is not immediately obvious from our theory that that would be the case. However, it would be interesting to check.

---

### Official Review · Reviewer_1ATq · 2025-04-01
**Solid theoretical contributions, yet somewhat limited empirical utility**

**Soundness:** 2
**Clarity:** 2

**Comments:**

The paper introduces a new, bias-corrected CKA estimator that addresses the shortcomings of existing methods by correcting for biases from both limited sampling of stimuli and neurons (a common situation in neuroscience). While existing debiased variants like that of Song et al. correct for stimulus sampling bias, this new approach mitigates the bias introduced by both limited neuron and stimulus sampling. The work is relevant to both the neuroscience and machine learning communities which both use CKA as a comon tool for representational comparisons.  The paper offers a solid theoretical contribution, though its empirical support for the practical utility of this metric remains somewhat limited.

Overall, the submission could be of high interest to the CCN audience, particularly because CKA is a widely used tool for representational comparisons and the neuron sampling issue is especially problematic in ephys data which are frequently used for inter-animal and model-to-brain comparisons. They also validate this estimator with various experiments. They first show the benefits on a synthetic dataset where the true alignment is perfect. In this case, their estimator recovers a CKA value of 1 even with sparse neuron sampling, whereas the Song et al. variant underestimates similarity due to its uncorrected bias from limited feature sampling. They also show when using the proposed debiased estimator, the model-to-brain alignment values not only follow the expected hierarchical pattern (with early model layers aligning with early visual areas like V1 and deeper layers with later areas like IT) as other estimators but also more closely match the brain-to-brain alignment benchmarks. However, the disentanglement experiments are less convincing. It remains unclear what additional insights these experiments provide over other estimators or metrics, which somewhat weakens the overall empirical validation in this area.

 In terms of clarity, the paper is well written and thorough in its theoretical exposition, though certain aspects (particularly the dense mathematical derivations) may be less accessible to a broader audience. Overall, the submission is both interesting and methodologically sound, despite some limitations in demonstrating the overall practical utility of this estimator in model-brain comparisons in terms of affording new insights.

**Expertise:**

2

**Interest:**

2

---

> ### Author Rebuttal · Authors · 2025-04-15
>
> We thank the reviewer for carefully reading our manuscript and providing valuable feedback.
>
> We would like to emphasize that there is a timely and important practical utility of our CKA estimator: measuring brain-to-brain alignment. Kohitij Kar and Jim DiCarlo, the major contributors in the field of model-to-brain comparisons (BrainScore), have recently highlighted the importance of measuring the brain-to-brain alignment as a benchmark for brain-to-model alignment at COSYNE 2025 conference last month (The abstract of their preliminary work: https://static1.squarespace.com/static/6102ca347474c263c40150cd/t/67e54c9db0e1c30043ea70ca/1743080615472/Cosyne2025_program_book.pdf#page=206.600). This shows that there is an urgent need for measuring the brain-to-brain alignment, yet a robust method is lacking. We would like to note that, although Kar and DiCarlo’s (unpublished) work provides an important insight into brain-to-brain similarity, their work relies heavily on heuristic extrapolation of sample-size sensitive similarity measures to overcome the finite sample effect, which is non-rigorous. In contrast, our estimator solves this problem with a statistically principled method, and it can be applied easily to the practical dataset. We will make this point clearer in the main text.
>
> In disentanglement experiments, we agree that the overall trends marginally change between biased and unbiased estimates. However, 1) the CKA rankings significantly change, 2) the real effect will be visible when the same experiments are made with different sample sizes. We included the disentanglement experiment as an additional example of a potential application of our estimator, not to present a new insight into the ventral stream processing. The naive CKA is sensitive to the number of images used for each supercategory, which can potentially give incorrect/biased insight.
>
> We will improve the accessibility of the mathematical aspect of our manuscript, with particular attention to improving the readability of the derivation in the appendix.

---

### Official Review · Reviewer_NEq6 · 2025-04-02

**Soundness:** 2
**Clarity:** 3

**Comments:**

### Interest
Evaluating and improving metrics for representational similarity between different neural networks have been a central issue within the community relevant to CCN, and thus the scope of the manuscript is very timely and important. The authors identify the feature subsampling bias in existing CKA and propose a correction to it. As a showcase of its application, the authors used the neural data collected in different cortical areas following the ventral visual stream and showed the change of disentanglement across the processing.


### Soundness

1.

Firstly, I must clarify that I did not have time to go through all the derivation details in the appendix. Nevertheless, I found the authors' identification of the limitation of the existing method technically solid and well-formed and I appreciate the clear explanation and quick insights on in which cases the biases exist and thus the correction is useful.

2.

The biggest question I have is about the core assumption of the bias being prominent, Eq.(8) and line 225, is when $\gamma \ll Q$, $\gamma$ being intrinsic dimensionalities of the underlying representation and $Q$ being feature sampling size (e.g. # neurons), and I am not fully convinced if this is a practical assumption. I believe that a core premise of neural representation and population coding is that there exists lower dimensional space which encodes the stimulus that population activity, and thus it is reasonable to say $\gamma$ is much smaller than the actual data dimension and often lies in very small dimension compare to neuronal collection. In essence, I think in there are many cases lie in a regime where this assumption does not hold.

That said,  I do not think this hurts the contribution of this paper; but rather it would strengthen the paper if the author could provide more ablations - for example I think providing the ablation of Figure. 2 right panel with assuming independent neurons (since Figure 2 with single neuron assumption I believe is the worst case biases and most of the interesting cases for representational similarity measurement concerns different sets of neurons/networks than strongly correlated ones ) might give us better sense how more prominent the biases becomes in more reasonable settings when the assumption does not hold.
 As minor questions, it is unclear to me what is 'Theory' value in the Figure 2, and could you provide a intuitive explanation why $CKA_{S}$ is worse than $CKA_{0}$?

3.

Another observation that I made is in both linear experiment and the actual neural data experiment, there seems to be a tendency that the bias is overcorrected with very small sampling size (figure2 and figure3). Can authors comment on this? This could be also the reason for very high CKA in figure 5 in some cases. I'm curious if this could be related to violating the assumption between intrinsic dimensionalities and sampling size.


### Clarity


I think the strongest point of the paper is the clarity.  It is clearly written with the problem statement and what is the contribution of the paper. The application of the method on the real data is also well explained. I think there can be more references added in the last section on disentanglement of the visual representation as I believe there exist prior works of similar spirit -measuring representational similarity in disentangled representation- such as Higgins et al. (2021), and probably authors could add some more references for the claim in the introduction around line 81.

**Expertise:**

2

**Interest:**

2

---

> ### Author Rebuttal · Authors · 2025-04-15
>
> We thank the reviewer for carefully reading our manuscript and providing valuable feedback.
>
> First, we address the reviewer’s concern regarding the soundness:
>
> 2. Regarding Eqn. 8 and $\gamma \gg Q$ case: we use Eqn. 8 to show that the existing CKA estimator $\widehat{CKA}_S$ has a small bias only when the dimensionality $\gamma$ is much smaller than the number of neurons sampled $Q$. Therefore, $\widehat{CKA}_S$ requires the $\gamma \ll Q$ assumption, whereas our estimator does not.
>
>     The reviewer points out that, in many real data, $\gamma$ should be small enough such that $\gamma \ll Q$, so $\widehat{CKA}_S$ is not significantly biased. Here we emphasize that one cannot make such an assumption: Stringer et al. 2019 [1] shows that the representation in the V1 is high-dimensional and the nth eigenvalue is $n^{-(\alpha+1)}$ where $\alpha$ is a small positive value (about $0.04$). In this case, $\gamma$ is  $\frac{6}{\pi^2\alpha^2}\approx 380$. This implies that even  $Q=1000$ is insufficient to reduce the bias in $\widehat{CKA}_S$. Our estimator does not require this.
>
>     We will provide a version of the Figure 2 right panel with a smaller number of neurons to show how the bias in  $\widehat{CKA}_S$ could get even worse.
>
>     The “Theory” lines (yellow dotted lines) in Fig. 2 are the theoretically predicted values of the estimators. This is currently explained in the figure caption, but we will make this point clearer.
>
>     **“Could you provide a intuitive explanation why**  $\widehat{CKA}_S$  **is worse than**  $\widehat{CKA}_0$ **?“**
>
>     On average,  $\widehat{CKA}_0$ overestimates  $\widehat{CKA}_S$, and   $\widehat{CKA}_S$ underestimates the true CKA. Therefore, sometimes, it is possible to observe $\widehat{CKA}_S$ being worse than $\widehat{CKA}_0$.
>
> 3. Our estimator can overestimate the true CKA in the very small sample size regime, as explained in detail in Appendix B.3 and noted in line 257 of the main text. We observe that our estimator $\widehat{CKA}_C$ is converged by the point where all relevant data is used, indicating there is no risk of overestimation in the  $\widehat{CKA}_C$  values reported in Fig. 5. We will include this analysis in the Appendix.
>
> In terms of clarity, we will add relevant works on object disentanglements as suggested by the reviewer.
>
> [1] Stringer et al., High-dimensional geometry of population responses in visual cortex, Nature 2019

---

> > ### Comment · Reviewer_NEq6 · 2025-04-21
> >
> > Thank you for the answers.
> > Regarding the discussion 2, I think the authors' point is adequate, but it is worth mentioning that there is still a lower dimensional semantic representation (such as categories, that authors use as disentanglement experiment).
> > Also, while the argument is specific to visual representation, I  acknowledge that lots of studies on representational similarity are focused on visual model which validates the relevance of the work and the assumption.
> >
> > Thank you again for your rebuttals and hope the feedback helped.
> > I believe my previous score already reflects the strength of the paper, and I'll keep my score.

---

### Meta-Review · Area_Chair_5RaU · 2025-05-06

**Ccn Recommendation:** Accept as Proceedings

**Metareview:**

There seems to be a general consensus amongst the three reviewers that the findings are interesting and timely. Reviewer 2 noted that the paper currently has only limited empirical support for the practical utility of this new bias-corrected CKA estimator.

Overall the assessment of the reviewers was positive, and after input from the SAC, we feel this manuscript merits acceptance. The fact that it is an AI based model which is linked to cognition and (blunting of) neurotransmitter processing makes it, in the opinion of the SAC (and PC) of broad interest to the community.

**Summary:**

All three reviewers indicated that both the soundness and clarity of the paper are at least adequate, with one reviewer rating the soundness as ‘strong’ and another rating the clarity as ‘exceptional.’ In addition, all reviewers indicated that the interest of the paper was ‘disciplinary’ with work relevant to both the neuroscience and machine learning communities. The paper was noted for being mathematically dense and the authors responded that they would improve the clarity of the derivations in the final version of the manuscript. As Reviewer 3 commented, and the AC agrees, the paper would benefit from more intuition building examples and sanity checks (similar to Figure 2).

**Expertise:**

2